# Stirred suspension bioreactors maintain naïve pluripotency of human pluripotent stem cells

Leili Rohani[1], Breanna S. Borys[2,3], Golsa Razian[1], Pooyan Naghsh[1], Shiying Liu[1], Adiv A. Johnson[4], Pranav Machiraju [5], Heidrun Holland[6], Ian A. Lewis[7], Ryan A. Groves[7], Derek Toms [8], Paul M. K. Gordon[9], Joyce W. Li[1], Tania So[2,3], Tiffany Dang[2,3], Michael S. Kallos[2,3] & Derrick E. Rancourt [1✉]

Due to their ability to standardize key physiological parameters, stirred suspension bioreactors can potentially scale the production of quality-controlled pluripotent stem cells (PSCs) for cell therapy application. Because of differences in bioreactor expansion efficiency between mouse (m) and human (h) PSCs, we investigated if conversion of hPSCs, from the conventional "primed" pluripotent state towards the "naïve" state prevalent in mPSCs, could be used to enhance hPSC production. Through transcriptomic enrichment of mechanosensing signaling, the expression of epigenetic regulators, metabolomics, and cell-surface protein marker analyses, we show that the stirred suspension bioreactor environment helps maintain a naïve-like pluripotent state. Our research corroborates that converting hPSCs towards a naïve state enhances hPSC manufacturing and indicates a potentially important role for the stirred suspension bioreactor's mechanical environment in maintaining naïve-like pluripotency.

[1] Department of Biochemistry and Molecular Biology, Cumming School of Medicine, University of Calgary, Calgary, AB, Canada. [2] Pharmaceutical Production Research Facility, Schulich School of Engineering, University of Calgary, Calgary, AB, Canada. [3] Biomedical Engineering Graduate Program, University of Calgary, Calgary, AB, Canada. [4] Nikon Instruments, Melville, New York, NY, USA. [5] Department of Paediatrics and Alberta Children's Hospital Research Institute, Cumming School of Medicine, University of Calgary, Calgary, AB, Canada. [6] Saxonian Incubator for Clinical Translation (SIKT), University of Leipzig, Leipzig, Germany. [7] Department of Biological Sciences, University of Calgary, Calgary, AB, Canada. [8] Department of Comparative Biology and Experimental Medicine, Faculty of Veterinary Medicine, University of Calgary, Calgary, AB, Canada. [9] CSM Center for Health Genomic and Informatics, University of Calgary, Calgary, AB, Canada. ✉email: rancourt@ucalgary.ca

Human pluripotent stem cells (hPSCs) harbor enormous potential for regenerative medicine and offer a unique opportunity to advance the development of cell-based therapies[1]. The development of hPSC biobanks[1], clinical trials[2], drug screening[3], and disease modeling[4] all demand robust stem cell manufacturing and expansion. Although the hPSC manufacturing field is evolving rapidly, a bottleneck is throttling clinical application: existing technologies are limited in their ability to efficiently scale production of clinically-viable stem cells[5].

Previous work suggests that expanding hPSCs as aggregates in stirred suspension bioreactors offers significant production advantages compared to static cell culture[6,7]. Key physiological parameters can be controlled and standardized in these platforms, allowing high-quality batches with large cell numbers[8]. Although we[8–10] and others[11] have demonstrated that murine (m) PSCs are expanded robustly in suspension bioreactors, hPSC expansion in these systems have resulted in lower fold-expansion and yields[12] and high costs[13–15]. Compared to mPSCs, the manufacturing of conventional hPSCs in stirred suspension bioreactors has proven more difficult and has resulted in significantly lower fold-expansion[14,16].

Mouse and human PSCs exhibit distinctive epigenetic features, self-renewal signaling requirements, and differentiation potential. Conventional hPSCs display a "primed" state of pluripotency similar to mouse epiblast stem cells derived from the post-implantation epiblast[17]. Conversely, mPSCs reside in the so-called "naïve" pluripotent state resembling the pre-implantation epiblast[18]. Prior reports have shown that specific culture conditions can either generate naïve hPSCs directly from embryos or convert primed hPSCs to the naïve state[19]. Compared to the primed state, naïve hPSCs have advantageous bioprocessing properties, including a higher single-cell cloning efficiency[20], an enhanced differentiation potential[21], an advanced adherent growth rate[19,22], bivalent metabolic activity[22,23], an altered transcriptome, and a hypomethylated epigenome[20,22,24,25]. Currently, the field has a limited understanding of how the hPSC state influences bioprocessing and whether this can resolve existing cell manufacturing gaps.

Recently, Lipsitz et al. reported that the conversion of hPSCs to an alternative pluripotent state enhanced suspension culture yield[26]. However, pluripotency maintenance of this "high-suspension-yield" state was dependant on ERK inhibitory cytokine withdrawal from suspension culture. This suggests that the cell's interaction with its mechanical environment and media components plays a critical role in maintaining naïve-like pluripotency within the bioreactor.

Considering that study[26], the robust expansion of mPSCs in bioreactors[8–11,16], and knowing that naïve hPSCs resemble mPSCs, we report here that cell-state conversion significantly enhances hPSC bioprocessing in stirred suspension culture. Since naïve hPSCs are metastable[27], we sought to assess the relationship between naïve pluripotency and bioreactors. Specifically, we aimed to identify whether or not the bioreactor mechanical environment influences the naïve pluripotency state. Through transcriptomics, assessing the expression of epigenetic regulators, metabolomics, and cell-surface protein marker analyses, we show that the bioreactor environment helps maintain the naïve pluripotent state. We observed enrichment of the mechano-sensing HIPPO signalling in the stirred suspension bioreactor, supporting the hypothesis that a mechanical environment plays an important role in maintaining naïve pluripotency. This observed positive influence of the bioreactor environment on naïve pluripotency indicates it may be possible to revert to naïve pluripotency in the bioreactor. Further optimization could enhance hPSC biomanufacturing for both clinical and research applications.

## Results

**Cell-state conversion enhances hPSC bioprocessing.** Previous reports have exploited combinations of small molecules, inhibitors, and genetic manipulation[20–22,28,29] to convert hPSCs from primed-to-naïve pluripotency. In our study, the conversion was achieved using commercial RSeT™ medium. This is a defined media based on a bFGF- and TGFβ-free version of NHSM[20]. Conversion in static culture produced robust cultures with naïve hPSC features, including tightly packed, domed colonies with refractive edges (Fig. 1). Resulting naïve hPSC colonies were maintained on inactivated mouse embryonic fibroblasts (iMEFs) in RSeT medium for several passages (Fig. 1). From passage (P) four onwards, the resulting established naïve hPSC colonies from static culture (P4) showed high expression levels of naïve pluripotency markers and the disappearance of *XIST* gene expression (Supplementary Fig. 1a), the latter being a hallmark of naïve pluripotency[30].

We have previously shown significant interaction effects between inoculation density and agitation rate for hPSCs in stirred suspension culture[31]. Here, we applied different inoculation cell densities and agitation rates for naïve hPSC in stirred suspension cultures. We found that uniform-sized aggregates formed at inoculation densities of 1E4, 2.5E4, and 5E4 cells/mL in the bioreactor (Supplementary Fig. 1b). We also observed that an agitation rate of 100 RPM (maximum fluid shear 6 dynes/cm$^2$) facilitated the formation of aggregates with a healthy morphology and an average diameter below levels where we would expect necrosis to occur[32]. We increased seeding densities of naïve hPSCs in suspension culture through agitated well plate systems. Naïve hPSCs were cultured in 24- and 6-well shaken suspension plates at seeding densities of 1E5 and 2E5 cells/mL for five days, and cultures underwent a single 60% media exchange on day 3. Unlike bioreactor-cultured, naïve hPSCs, shaken suspension culture in well-plates resulted in less uniform aggregate sizes with dark centers that may represent partly necrotic or apoptotic cores (Supplementary Fig. 1c).

Having identified a successful combination of inoculation cell density and agitation rate in stirred suspension bioreactors, we assessed growth kinetics and aggregate formation characteristics in both naïve and primed hPSCs. To evaluate each cell type's growth kinetics in stirred suspension culture, newly-converted naïve hPSCs (P0 from static culture), established naïve hPSCs (P4 from static culture), and equivalent primed hPSCs, were each inoculated into separate 100 mL stirred suspension bioreactors agitated at 100 rpm. Naïve hPSCs were then counted at passages one (P1) and five (P5), respectively. Batch conditions were used to determine growth kinetics for five days post-inoculation without the effect of a feeding regime. Here, growth rates and multiplication ratios were significantly greater, while exponential doubling times were significantly shorter for both P1 and P5 naïve hPSCs compared to their primed counterparts (Fig. 2a, Supplementary Fig. 2a). The exponential growth rates for P1 naïve hPSCs) and P5 naïve hPSCs were 1.5X greater than those observed in primed hPSCs.

In agreement with previous studies, primed hPSCs experienced extensive cell death upon inoculation and an extended lag period in bioreactor culture[12,26,33]. By contrast, naïve hPSCs recovered and expanded in unoptimized batch culture conditions with fold-increases of $4.09 \pm 0.09$ (P1) to $6.90 \pm 0.03$ (P5) within five days (Fig. 2a and Supplementary Fig. 2a). Exponential growth rates were comparable between P1 and P5 naïve hPSCs, but the fold-increase was greater for P5 (Fig. 2a). The difference in fold-increase between P1 and P5 was concomitant with the disappearance of *XIST* gene expression with subsequent passages (Supplementary Fig. 1a).

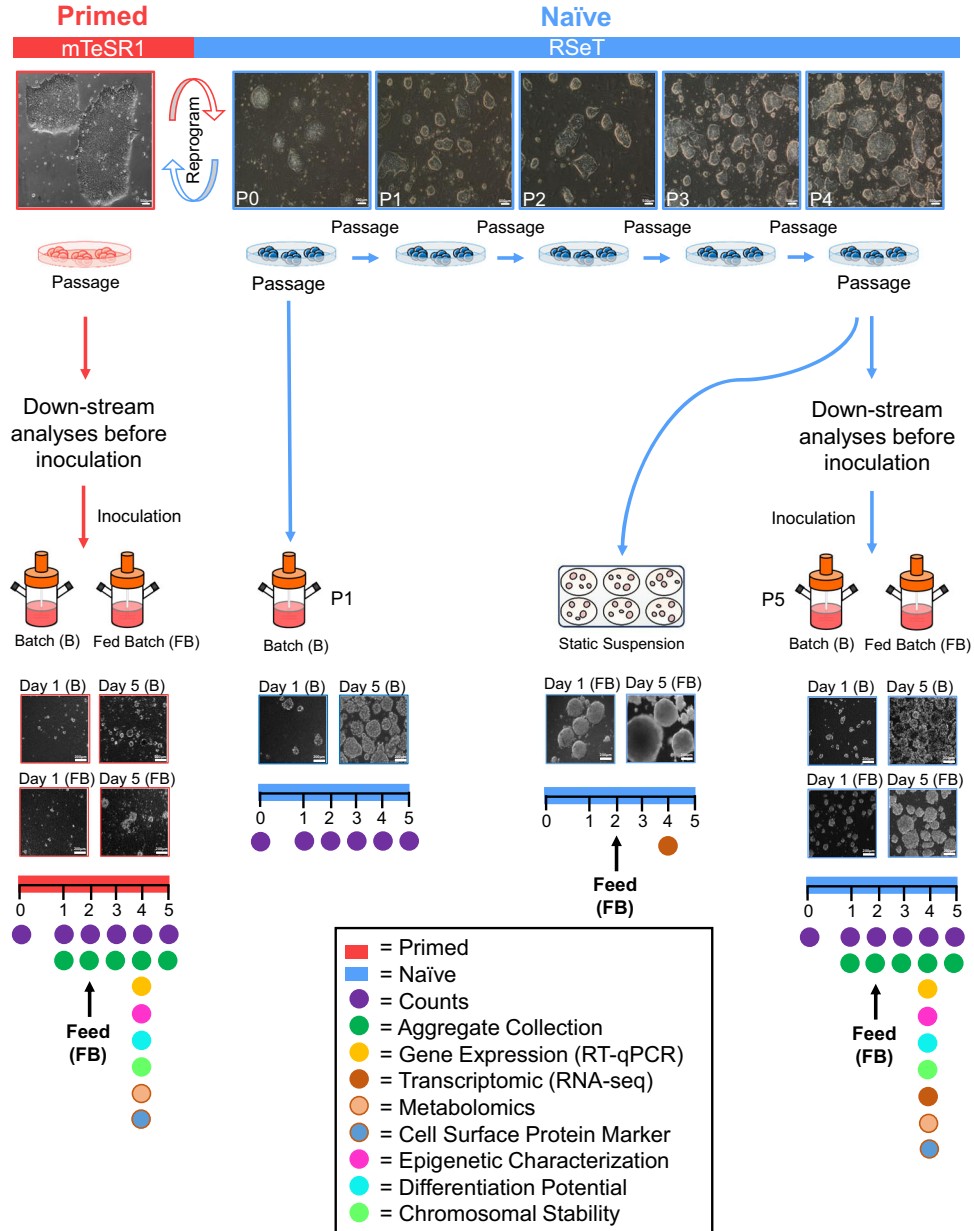

**Fig. 1 Experimental design. Conversion from a primed to naïve state of pluripotency.** Schematic representation of experimental design and morphological changes during transition from primed (flat) towards naïve (dome-shaped) hPSC colonies in static culture (upper panel). Conversion from the primed to naïve pluripotent state was done using RSeT media. Naïve hPSC colonies are shown at several passages (P0–P4) on iMEFs following conversion (upper panel). The arrowheads under the static culture dishes indicate the inoculation of cells from static culture into static suspension and stirred suspension culture for each naïve (P0 and P4) and primed hPSC sample. Following inoculation, naïve hPSCs were counted as passages one and five, respectively. Aliquots of static-cultured cells from naïve (P4) and primed hPSC samples were collected for downstream analyses before inoculation. Batch and fed-batch conditions were used for suspension culture in bioreactors. A fed-batch condition was used for the static suspension culture condition. Sample images of aggregate morphology for naïve and primed hPSCs at days 1 and 5 post-inoculation under batch and fed-batch are shown. The bars below the images of an aggregate's morphology demonstrate the number of days post-inoculation. Static suspension and stirred suspension bioreactor sample collections for various analyses are shown by different coloured-bullets in the lower-most panel. The days of sample collections for different analyses are shown by the coloured-bullets under the bar. For the fed-batch condition, the media change and feeding were done on day 2 post-inoculation (60% media change, 48-h post-inoculation). Scale bars = 200 μm. iMEFs inactivated mouse embryonic fibroblasts.

To evaluate the effect of feeding regime on growth kinetics in stirred suspension culture, established naïve (P4 from static culture) and equivalent primed hPSCs were inoculated into bioreactors under batch and fed-batch conditions (Fig. 2b). We used established naïve hPSCs (P4 from static culture) as they demonstrated a superior fold-increase compared to newly-formed naïve hPSCs (P0 from static culture) (Fig. 2a). Based on our previous work[31], we examined the effect of a 60% media exchange 48 h after cell inoculation. For both primed and naïve hPSCs, there was no significant difference in cell expansion or growth kinetics between batch and fed-batch conditions. This indicates that nutrient limitations or buildup of metabolites and waste

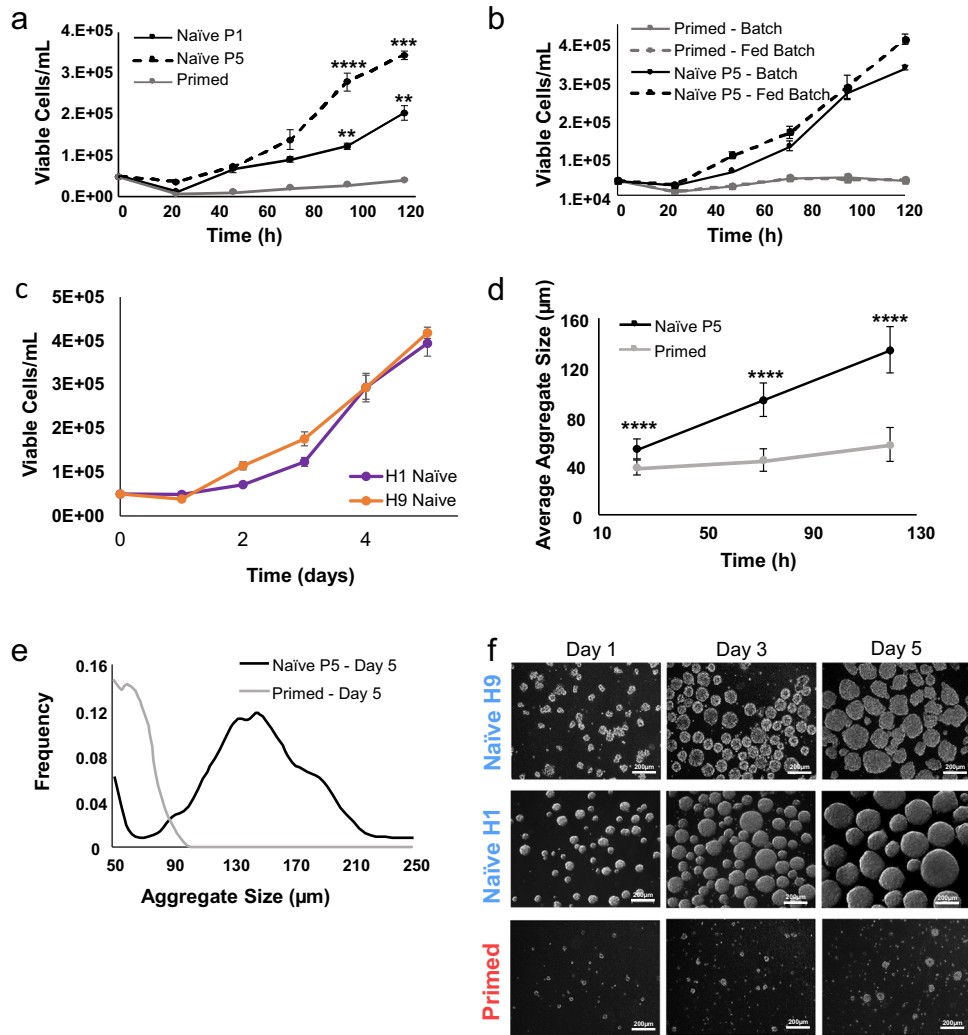

**Fig. 2 Bioreactor growth kinetics of naïve and primed hPSCs. a** Growth kinetics of naïve and primed hPSCs at different days (D0–D5) post-inoculation under batch condition. Statically cultured, naïve hPSCs were inoculated into bioreactors at passages zero and four following primed to naïve conversion in static culture wherein, they were counted as passages one and five, respectively. **b** Growth kinetics of naïve and primed hPSCs at different days (D0–D5) post-inoculation under batch and fed-batch (60% media change, 48-h post-inoculation) conditions. Static-cultured, naïve hPSCs were inoculated into bioreactors at passage four following conversion in static culture, wherein they were counted as passage five. **c** Growth kinetics of naïve hPSCs from H1 and H9 hESC lines at different days (D0–D5) post-inoculation under fed-batch (60% media change, 48-h post-inoculation) conditions. Static-cultured, naïve hPSCs for both cell lines were inoculated into bioreactors at passage four following conversion in static culture, wherein they were counted as passage five. **d** Bioreactor-cultured aggregate growth. Average aggregate size was measured on days 1–5 post-inoculation to detect bioreactor-cultured aggregate growth in naïve (P5) and primed hPSC samples. **e** Size distribution of bioreactor-cultured aggregates. Distribution of bioreactor-cultured, naïve (P5) and primed hPSC aggregates were measured on day 5 post-inoculation using the Beckman Coulter Multisizer III. It was defined as multi-cellular spheroids with a mean diameter greater than 50 µm. **f** Aggregate growth morphology. Representative images show naïve and primed hPSC aggregate morphology at different days (D1, D3, and D5) post-inoculation. The data presented are generated from inoculation density of 50,000 cells/mL. The data are represented as mean ± SEM ($n = 4$). **** = Adj-P < 0.0001 using GraphPad Prism. Scale bars = 200 µm.

products were likely not limiting factors for cell growth. Under both batch and fed-batch conditions, bioreactor-cultured, naïve hPSCs showed significantly superior growth kinetics compared to their primed equivalent (Fig. 2b). Both H1 and H9 naïve hPSC lines showed comparable average fold-expansion (7.88 and 8.37, respectively) following five days of fed-batch stirred suspension culture (Fig. 2c).

Naïve hPSCs seeded at higher densities in shaken suspension well-plates (1E5 and 2E5 cells/mL) exhibited similar day 3 fold-expansions (1.78 and 2.73-fold, respectively) to the bioreactor culture seeded at the lower density of 5E4 cells/mL (2.47-fold). After day 3, however, cell growth in the shaken suspension well-plates plateaued (Supplementary Fig. 2b). The expansion plateau

at this point could be due to the feeding strategy and might indicate that more nutrients may be required for continued expansion of naïve hPSCs seeded at higher densities in the shaken suspension well-plates.

We next evaluated the aggregate formation characteristics of naïve and primed hPSCs in suspension culture, as aggregate size is an important bioprocessing variable that has implications on cell health and differentiation[32]. We assessed aggregate growth by measuring the average aggregate size for five days post-inoculation in samples from naïve (P5) and equivalent primed hPSCs in stirred suspension culture. Naïve hPSC aggregates showed significantly greater growth compared to their primed counterparts at all time points (Fig. 2d). Primed hPSC aggregates

that formed on day 1 had an average diameter of 38.8 ± 6.5 μm and expanded minimally, reaching an average diameter of only 57.3 ± 13.7 μm by day 5. Aggregates of naïve hPSCs increased from an average diameter of 54.0 ± 6.5 μm on day 1 to 134.4 ± 18.6 μm on day 5 (Fig. 2d). The naïve hPSC aggregates had a relatively uniform size distribution showing a single defined peak at an average diameter of 130–200 μm only four days following bioreactor seeding, which is an ideal aggregate size based on our data and data from others[31,34]. By contrast, primed hPSC aggregates had a larger standard deviation in diameter size in the range of 50–90 μm (Fig. 2e). Primed hPSCs showed poor aggregation and were smaller than naïve hPSC aggregates, which showed few single cells one day post-inoculation, and further formed larger uniform aggregates past days 3 and 5 in stirred suspension bioreactors (Fig. 2f).

We further attempted to serial passage naïve hPSCs in 6-well shaken suspension plates (seeded at 2E5 cells/mL, high density) in addition to 100 mL stirred suspension bioreactors (seeded at 5E4 cells/mL, low density). The cells were passaged every five days. In both systems, following the first few days of passage two, the aggregates began to fall apart (Supplementary Figs. 2c, d). In the first three days of passaging, percent viability declined from 52 ± 1.65% on day 1 to 39 ± 2.2% on day 3 (Supplementary Fig. 2e). Although aggregates still were able to form (Supplementary Fig. 2f), the growth was reduced with each passage.

**Bioreactors support a naïve pluripotency transcriptomic state.** In-line with previous naïve hPSC reports[24–26,28,29,35], our RT-qPCR analysis showed elevated expression levels of naïve hPSC-associated genes (i.e., KLF2 and KLF5) in bioreactor-cultured, naïve hPSCs (Fig. 3a). Consistent with previously published findings[24,25,35,36], we also observed significantly reduced expression of the primed hPSC-associated genes (i.e., OTX2, ZIC2, and DUSP6) in bioreactor-cultured, naïve hPSCs (Fig. 3a, Supplementary Fig. 3a).

Increased levels of naïve (i.e., KLF2, KLF4, and STELLA) and decreased levels of primed (i.e., CD24) pluripotency genes were observed in naïve hPSCs cultured in the bioreactor (Supplementary Fig. 3b). We also saw an intermediate level of KLF4 expression in bioreactor-cultured, naïve hPSCs, which is consistent with previous reports showing that KLF4 is intermediately expressed in naïve hPSCs converted by RSeT medium[35,36]. Regardless, the expression level of KLF4 was high in bioreactor-cultured, naïve hPSCs compared to their primed counterparts (Supplementary Fig. 3b). Our finding that bioreactor-cultured, naïve hPSCs display strong KLF2 expression (Supplementary Fig. 3b) corroborates previous data favoring an essential role for KLF2 in hPSC conversion to the naïve pluripotent state[28].

We further examined the transcriptome with RNA-sequencing (RNA-seq) to identify whether the bioreactor environment is particularly involved in naïve pluripotency maintenance. For this analysis, we designed experiments to compare naïve hPSCs cultured under three different conditions: (1) static; (2) static suspension; and (3) stirred suspension culture. Principal component analysis (PCA) revealed that cells from each culture condition clustered together as distinct groups separate from the other conditions (Supplementary Fig. 3c). Moreover, PCA revealed two major components: the first discriminating between the cells from static suspension and stirred suspension culture with 66% of the observed variance (PC1), and the second discriminating between static suspension and static culture conditions with 16% of the variance (PC2) (Supplementary Fig. 3c). The cells from static suspension formed an intermediate cluster, tended to be more like the static condition (PC2), and

preserved more distance from the stirred suspension culture (PC1) (Supplementary Fig. 3c).

Transcriptomic comparison of naïve hPSCs cultured under static suspension and static conditions revealed that 4,039 transcripts were differentially expressed (Supplementary Data 1). We used Ingenuity Pathway Analysis (IPA) and its z-score algorithm to identify enriched canonical pathways that were more or less active (positive or negative z-score) according to the IPA database. To enhance the stringency of our analysis, we considered z-score ≥2 or ≤−2, and –log10(Benjamini–Hochberg multiple testing corrected p value of differential expression) ≥2 as the filter criteria for pathway enrichment significance. The results of IPA showed enrichment of 9 canonical pathways in static suspension compared to static condition (Supplementary Data 2). We found negative enrichment (negative z-score, implying downregulation) of the Mouse ESC Pluripotency pathway (z-score: −2.302) in naïve hPSCs cultured in static suspension culture, corroborating the decrease of pluripotency in these cells (Fig. 3b, Supplementary Table 1). Furthermore, negative enrichment of pathways related to cell–cell connection and adhesion such as Actin Cytoskeleton and RhoA signalling were shown in static suspension culture (Fig. 3b, Supplementary Table 1). Positive enrichment (positive z-score, implying upregulation) of AMPK signaling (Fig. 3b, Supplementary Table 1) matched our observations that aggregates formed in static suspension condition demonstrated necrotic or apoptotic cores (Supplementary Fig. 1c). Previous reports showed that AMPK is activated in response to stressors such as ischemia and necrotic cores[37].

Transcriptomic comparison of naïve hPSCs cultured under stirred suspension and static culture revealed that 10,181 transcripts were differentially expressed (Supplementary Data 3). IPA analysis showed enrichment of 31 canonical pathways in stirred suspension compared to static culture (Supplementary Data 4). Substantially, we found positive enrichment of the mechano-sensing HIPPO signaling pathway in stirred suspension culture (z-score: 3.162) (Fig. 4a, Supplementary Table 2). We also found negative enrichment for several lineage specific pathways with neural bias such as Synaptogenesis as well as primed pluripotency related pathways such as Stearate Biosynthesis, in stirred suspension culture (Fig. 4a, Supplementary Table 2). Like static suspension culture, pathways related to cell–cell connection, adhesion, and extracellular matrix (ECM) such as Integrin exhibited negative enrichment in stirred suspension culture (Fig. 4a, Supplementary Table 2). More enriched canonical pathways are shown in Supplementary Data 4 and Supplementary Table 2.

Finally, we compared the transcriptomes of naïve hPSCs cultured under stirred suspension and static suspension culture and found that 8,938 transcripts were differentially expressed (Supplementary Data 5). IPA analysis showed enrichment of 34 canonical signalling pathways in stirred suspension compared to static suspension culture (Supplementary Data 6). Positive enrichment of the HIPPO signaling pathway in stirred suspension culture (z-score: 2.186) was shown (similar to stirred suspension and static culture pairwise comparison) (Fig. 4b, Supplementary Table 3), suggesting the appearance of mechano-sensing HIPPO signalling exclusively in stirred suspension culture. Interestingly, we also found positive enrichment of germ cell-related PTEN signaling pathway[38] in stirred suspension (z-score: 3.457) compared to static suspension culture (Fig. 4b, Supplementary Table 3). Furthermore, negative enrichment of primed pluripotency related pathways[13,20–22,39–41], such as ERK-MAPK and Cholesterol Biosynthesis was shown in stirred suspension culture. Like stirred suspension versus static culture pairwise comparison, we observed negative enrichment of multiple lineage and differentiation related pathways in stirred suspension compared

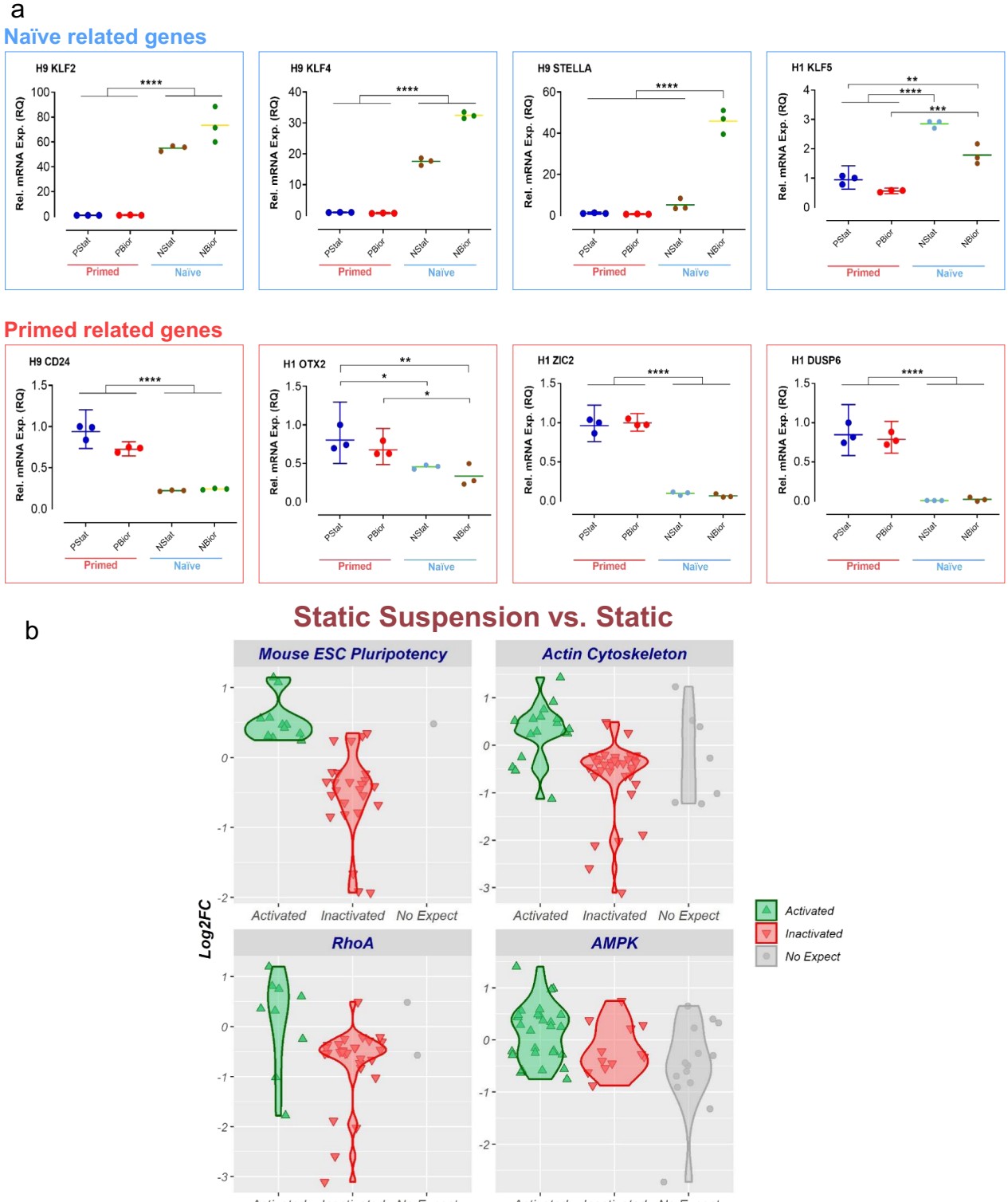

**Static Suspension vs. Static**

*Actual (experiment) Vs. expected (IPA) gene differential expression for canonical pathways*

to static suspension culture (Fig. 4b, Supplementary Table 3). Other enriched canonical pathways are detailed in Supplementary Data 6 and Supplementary Table 3.

Gene ontology (GO) analysis for our transcriptomic comparisons revealed enrichment of GO biological process terms linked to cellular component organization or biogenesis, cell–cell adhesion, macromolecular complex subunit organization, and

metabolic process in all three pairwise comparisons (GO accessions 0071840, 0098609, 0043933, and 0008152; Supplementary Fig. 4). This suggests the restructuring of cells in static suspension and stirred suspension relative to static culture. Cells cultured under static suspension exhibited enrichment related to the GO category "developmental processes" (GO:0032502), suggesting a greater differentiation state[20,24]. This observation is

**Fig. 3 Gene expression and transcriptomic analyses of naïve and primed hPSCs. a** Depiction of naïve- (KLF2, KLF4, STELLA, and KLF5) and primed- (CD24, OTX2, ZIC2, and DUSP6) hPSC-associated gene expression levels for H1 and H9 hESC lines analyzed by RT-qPCR. The aliquots of statically cultured, cells for each naïve (P4) and primed hPSC sample were collected for gene expression analysis before inoculating them into stirred suspension bioreactors. Those cells were counted as "NStat" and "PStat". The aggregates of day 4 post-inoculation were collected for each naïve and primed hPSC sample for gene expression analysis. Expression was quantified relative to the housekeeping gene GAPDH and was normalized to statically cultured, primed hPSC (PStat) level (=1). All the cultures underwent fed-batch condition (60% media change, 48-h post-inoculation) for bioreactor cultures. The data presented are generated from inoculation density of 50,000 cells/mL. The data are represented as mean ± SEM ($n = 3$). **** = Adj-P < 0.0001, *** = Adj-P = 0.0002 and 0.0004, ** = Adj-P = 0.0021, * = Adj-P = 0.0291 using GraphPad Prism. Adj-P = Adjusted $p$ value. RQ relative quantification. PStat = primed hPSCs cultured under static condition. PBior = primed hPSCs cultured under stirred suspension bioreactor condition. NStat = naïve hPSCs cultured under static condition. NBior = naïve hPSCs cultured under stirred suspension bioreactor condition. **b** RNA-sequencing analysis of naïve H9 hPSCs cultured under static suspension and static conditions. The violin plots show the enriched canonical pathways in static suspension compared to static culture (selected as a control reference) based on Ingenuity Pathway Analysis (IPA). The IPA z-score algorithm was used to identify enriched canonical pathways that are more active (positive z-score) or less active (negative z-score) according to the IPA database and observed gene expression in our RNA-seq dataset. Activated = genes expected to be upregulated (positive log2FC) if the pathway is activated according to IPA canonical pathway database. Inactivated = genes expected to be downregulated (negative log2FC) if the pathway is activated according to IPA canonical pathway database. No-expect = no direction of change expectation is available in the IPA canonical pathway database. Filter criteria for pathway enrichment significance was expected direction of change Z-score ≥ 2 or ≤−2, and –log10 (Benjamini–Hochberg multiple testing corrected $p$ value of differential expression) ≥2. The suspension culture underwent fed-batch condition (60% media change, 48-h post-inoculation). Aggregates of day 4 post-culture were collected for the analysis. The data are representative of three replicates.

concordant with IPA analysis showing downregulation of Mouse Embryonic Stem Cell Pluripotency pathway in static suspension culture (Supplementary Fig. 4 and Fig. 3b).

The cells cultured under stirred suspension condition (versus both static and static suspension) were highly enriched for GO terms linked to cellular localization, organelle organization, cell-cycle process, biosynthetic process, ribonucleoprotein complex biogenesis, mitochondrion organization, and several transcription-translation machinery related terms, such as gene expression, RNA processing, and ribosome biogenesis (Supplementary Fig. 4.). Many of these terms, such as mitotic cell cycle, RNA processing, and ribosome biogenesis, have already been reported in other studies related to the naïve pluripotency state[24,42]. Interestingly, the "cellular response to stress" term showed higher enrichment in stirred suspension culture, while enrichment of the term "intracellular transport" was only observed in stirred suspension culture (Supplementary Fig. 4). Additional GO terms from biological process (BP), cellular component (CC) and molecular function (MF) categories from all three pairwise comparisons are listed in Supplementary Data 7–9.

**Expression of naïve-related epigenetic transcripts and X-chromosome reactivation in bioreactors.** Previous reports have shown different patterns of expression for epigenetic regulatory transcripts in naïve and primed hPSCs in static culture[20,21,35,36]. We were interested to see what the expression level of the epigenetic regulatory transcripts would be in naïve and primed hPSCs cultured in stirred suspension bioreactors (Fig. 5a). The epigenetic regulatory transcripts, *DNMT3L*, *MAGEB2*, and *PRDM14*, were all significantly elevated in bioreactor-cultured, naïve hPSCs compared to their primed counterparts, while *DNMT1*, *TDG*, and *EP300* showed no significant expression differences (Fig. 5a, Supplementary Fig. 5a). While, there was no difference in the expression of *DNMT3B* between bioreactor-cultured, naïve and primed hPSCs for H1, the expression of *DNMT3B* was significantly upregulated in bioreactor-cultured, naïve H9 hPSCs compared to statically cultured, primed H9 hPSCs (Supplementary Fig. 5b).

The presence of two active X-chromosomes in female hPSCs is a hallmark of naïve pluripotency[43]. Conversely, primed hPSCs show an inactive X (Xi) chromosome[44]. We analyzed the intensity and distribution of H3K27me3, a marker of the Xi-chromosome[22], in the nuclei of bioreactor-cultured hPSC aggregates (Fig. 5b and Supplementary Fig. 5c). H3K27me3 was

fainter in the nuclei of naïve hPSC aggregates compared to primed hPSC aggregates (Fig. 5b and Supplementary Fig. 5c). Accumulation of condensed H3K27me3 foci in the nucleus is a classical sign of Xi-chromosome[22,30,35] and, as predicted, analyzed H3K27me3 foci (279 foci for naïve and 281 foci for primed) showed greater condensed foci in the nuclei of primed hPSC aggregates compared to their naïve counterparts (Fig. 5b). Accordingly, the H3K27me3 distribution plot revealed higher intensity and average percentage of condensed foci in the nuclei of primed hPSC aggregates (48%) relative to the nuclei of their naïve counterpart (0.3%) (Fig. 5b). The intensity and distribution of more H3K27me3 foci on six more nuclei in bioreactor-cultured, naïve and primed hPSC aggregates are illustrated in Supplementary Fig 4b. Confocal Z-stack videos of the stained aggregates for H3K27me3 are shown in Supplementary Movies 1–2. These results suggest that bioreactor-cultured, primed hPSC aggregates harbor an Xi-chromosome, while the X-chromosomes in corresponding naïve hPSC aggregates are reactivated.

**Robust production of naïve-related metabolites in bioreactors.** Naïve and primed hPSCs show distinct patterns of metabolism, which influence their epigenetic and pluripotency state[39,45,46]. We used high-resolution LC-MS semi-targeted metabolomics to measure extracellular metabolites in the media of bioreactor-cultured, naïve and primed hPSCs. Metabolites, such as α-ketoglutarate (α-KG), L-alanine, aspartate, lactate, glutamate, and malate, had distinguishable patterns between naïve and primed hPSCs in the bioreactor (Supplementary Data 10).

We tracked metabolite dynamics during the primed to naïve conversion of H1 hESCs in the bioreactor (Fig. 6). We gradually switched from mTESR1 to RSeT media (i.e., primed-to-naïve) over six days of culture (D2 to D6) and the media was collected every day for analysis. The dynamics of metabolites showed a gradual increase of α-KG and L-alanine from days 2 to 6 (Fig. 6). Levels of aspartate, glutamate, and lactate dropped from day 2 to 3, and then gradually declined from days 3 to 6 (Fig. 6). In contrast, malate showed a minor decrease from day 2 to 6 of culture.

We also screened extracellular metabolites in the media of H9 primed and established naïve (P5) hPSCs in the bioreactor. Like the H1 cells gradually converted in the bioreactor (above), high levels of α-KG and L-alanine were shown in established naïve (P5) compared to primed hPSCs cultured in the bioreactor (Fig. 6). Although, the H1 gradually converted cells showed a

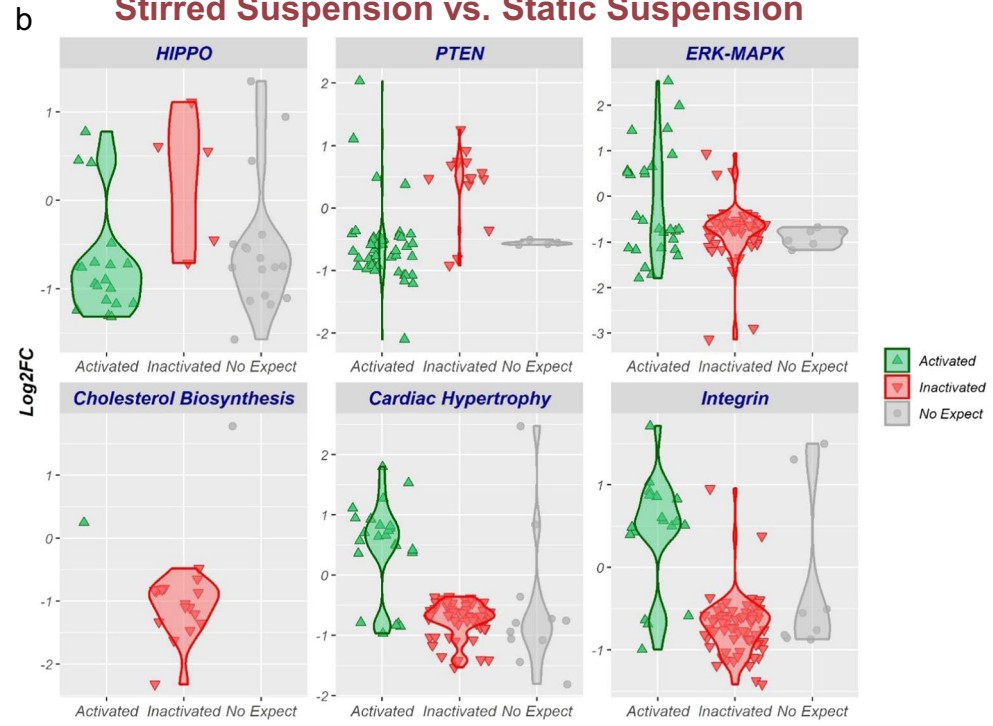

a **Stirred Suspension vs. Static**

Actual (experiment) Vs. expected (IPA) gene differential expression
for canonical pathways

b **Stirred Suspension vs. Static Suspension**

Actual (experiment) Vs. expected (IPA) gene differential expression for canonical pathways

decreasing pattern of glutamate from day 2 to 6, the level of glutamate was greatly increased in established naïve hPSCs compared to their primed counterpart (Fig. 6). Aspartate levels underwent a significant drop in naïve hPSCs in the bioreactor, while levels of lactate and malate were comparable in naïve and primed hPSCs in the bioreactor, with a tendency for malate to be increased in the media of naïve hPSCs (Fig. 6).

**Naïve hPSCs downregulate primed-related surface markers**. Specific cell-surface proteins were previously utilized to track the dynamics of hPSCs during primed-naïve conversions[36,47,48], allowing for the characterization of naïve hPSCs. Here, we used a similar approach to characterize our bioreactor-cultured, naïve and primed hPSCs. Using flow cytometry, we tracked the expression of CD75/SUSD2 and CD90/CD24/CD317

**Fig. 4 Transcriptomic analysis of naïve hPSCs cultured under static, static suspension, and stirred suspension culture. a** RNA-sequencing analysis of naïve H9 hPSCs cultured under stirred suspension and static culture. **b** RNA-sequencing analysis of naïve H9 hPSCs cultured under stirred suspension and static suspension culture. The violin plots show the enriched canonical pathways in the stirred suspension compared to static and static suspension cultures (selected as control references) based on ingenuity pathway analysis (IPA). The IPA z-score algorithm was used to identify enriched canonical pathways that are more active (positive z-score) or less active (negative z-score) according to the IPA database and observed gene expression in our RNA-seq dataset. Activated = genes expected to be upregulated (positive log2FC) if the pathway is activated according to IPA canonical pathway database. Inactivated = genes expected to be downregulated (negative log2FC) if the pathway is activated according to IPA canonical pathway database. No-expect = no direction of change expectation is available in the IPA canonical pathway database. Filter criteria for pathway enrichment significance was expected direction of change Z-score $\geq 2$ or $\leq -2$, and –log10 (Benjamini–Hochberg multiple testing corrected $p$ value of differential expression) $\geq 2$. All the suspension cultures underwent fed-batch condition (60% media change, 48-h post-inoculation). Aggregates of day 4 post-cultures were collected for the analysis. The data presented are generated from bioreactor inoculation density of 50,000 cells/mL. The data are representative of three replicates.

as naïve and primed hPSC-specific cell-surface markers, respectively[25,47–49]. We observed no significant differences in the expression of CD75 or SUSD2 between naïve and primed hPSCs in the bioreactor (Fig. 7a–c), although histograms revealed a separation in the fluorescence signal between naïve and primed hPSCs in the bioreactor with wider distribution for bioreactor-cultured, naïve hPSCs (Fig. 7c). Similarly, pairwise comparison between bioreactor-cultured, naïve hPSCs and their statically cultured, naïve counterparts showed no remarkable difference for CD75 or SUSD2 expression, although this revealed a separation in histograms (Supplementary Fig. 6a, c). There was no separation in fluorescent signal of CD75 or SUSD2 in the histogram comparing bioreactor-cultured, primed hPSCs and their statically cultured, primed counterparts (Supplementary Fig. 6b, d). We also tracked the SUSD2 dynamics during primed to naïve conversion in the bioreactor in a time-course (mentioned above for metabolomics) where the cells were collected every day from day 2 to 6 for analysis. The results revealed that the expression dynamic of SUSD2 remained stable without a significant shift in six days of bioreactor culture (Fig. 7d).

Conversely, compared to their primed equivalents, bioreactor-cultured, naïve hPSCs revealed reduced expression of all primed hPSC-specific markers: CD90, CD24 and CD317. Likewise, expression histograms indicated lower expression of those surface markers in bioreactor-cultured, naïve hPSCs, with a wider distribution particularly for CD317 (Fig. 8a–c, Supplementary Fig. 7a). Pairwise comparison of naïve hPSCs cultured in static or bioreactor revealed no distinctive expression of CD24 yet showed a propensity for less expression of CD90 and CD317 in the bioreactor with wider distribution particularly for CD317 (Supplementary Fig. 7b, d, e). No distinction was found for CD90 and CD317 between primed hPSCs cultured statically or in the bioreactor (Supplementary Fig. 7c, f). We also tracked the expression dynamics of CD24 during primed to naïve conversion in the bioreactor (Fig. 8d). Within six days of culture, CD24 expression exhibited a different dynamic compared to SUSD2, which persisted until day 5 before undergoing downregulation on day 6 (Fig. 8d). Analyses of CD75 and CD90 through flow cytometry dot plots (including gating strategy) in both bioreactor-cultured, naïve and primed hPSCs are summarized in Supplementary Figs. 6e and 8.

**Naïve hPSCs show multi-lineage potency and stable chromosomes.** We further aimed to evaluate the expression and localization of pluripotency- and naïve-specific protein markers in bioreactor-cultured, naïve and primed hPSC aggregates. As expected, both bioreactor-cultured, naïve and primed hPSC aggregates showed high nuclear expression of the pluripotency marker OCT4 (Fig. 9). The expression of STELLA was similar in both naïve and primed hPSC aggregates (Fig. 9). In line with previous reports on the characteristics of the naïve pluripotent state[20,50], both TFE3 and KLF4 were translocated into the nuclei of bioreactor-cultured, naïve hPSC aggregates (Fig. 9). By

contrast, bioreactor-cultured, primed hPSC aggregates showed predominantly cytoplasmic localization of TFE3. However, some nuclear translocation of TFE3 was observed in the outer layer of bioreactor-cultured, primed hPSC aggregates (Supplementary Fig. 9a). The outer layer of hPSC aggregates are directly exposed to fluid shear stress within the bioreactor, this direct force might help the nuclear translocation of TFE3 in the outer layer of primed hPSC aggregates. Confocal Z-stack videos of the whole-mount immuno-stained aggregates are shown in Supplementary Movies 3–6.

We next investigated the differentiation potential of bioreactor-cultured, naïve hPSCs by studying teratoma formation and targeted in vitro differentiation. Bioreactor-cultured, H9 naïve hPSCs formed teratomas after only 3 weeks post-transplantation into SCID mice. In sharp contrast, primed hPSC counterparts formed teratomas 8–12 weeks following transplantation. Cell types related to all three germ layers were formed in teratomas generated from both bioreactor-cultured, naïve and primed hPSCs (Fig. 10a). Naïve hPSCs showed a higher propensity for differentiation into the endoderm lineage and a reduced capacity for mesoderm lineage differentiation (Supplementary Fig. 9b), as has been previously reported[29,51].

Bioreactor-cultured, naïve hPSCs from H1 hESC line could also be directly differentiated into cells representing all three germ layers: cardiomyocytes (mesoderm), hepatocytes (endoderm), and neural rosettes (ectoderm) in vitro. This was done using previously published protocols for each specific differentiation[26,52,53]. Beating clusters of cardiomyocytes (Supplementary Movie 7) were purified using a lactate-based metabolic selection and, at day 20, mature cardiomyocytes showed robust expression of TNNT2 with clear appearance of linear Z-disk organocations (Fig. 10b). Hepatocytes derived from bioreactor-cultured, naïve hPSCs exhibited nuclear expression of HNF4α and cytoplasmic peri-nuclear expression of CYP3A4 (Fig. 10b). Bioreactor-cultured, naïve hPSCs also efficiently differentiated into ectodermal neural rosettes containing columnar cells expressing nuclear PAX6 (Fig. 10b).

A major concern for the conversion of hPSCs into the naïve state is the potential for selecting genetic aberrations or chromosomal instability[28,35,54,55]. We used spectral karyotyping (SKY) to assess, in detail, the genomic health of bioreactor-cultured, naïve and primed hPSCs (Fig. 10c, Supplementary Fig. 10a, b).

We found that naïve and primed hPSCs grown in bioreactors had stable karyotypes without clonal occurrence of structural chromosomal aberrations (Supplementary Fig. 10). No further structural chromosomal aberrations were observed in bioreactor-cultured, naïve hPSCs. However, bioreactor-cultured, primed hPSCs revealed a structural chromosomal aberration as a single event, as seen by an unbalanced nonreciprocal translocation between chromosomes 19 and 20 (Supplementary Fig. 10a, b). We also evaluated structural chromosomal aberrations in bioreactor-cultured, naïve and primed hPSCs following recovery

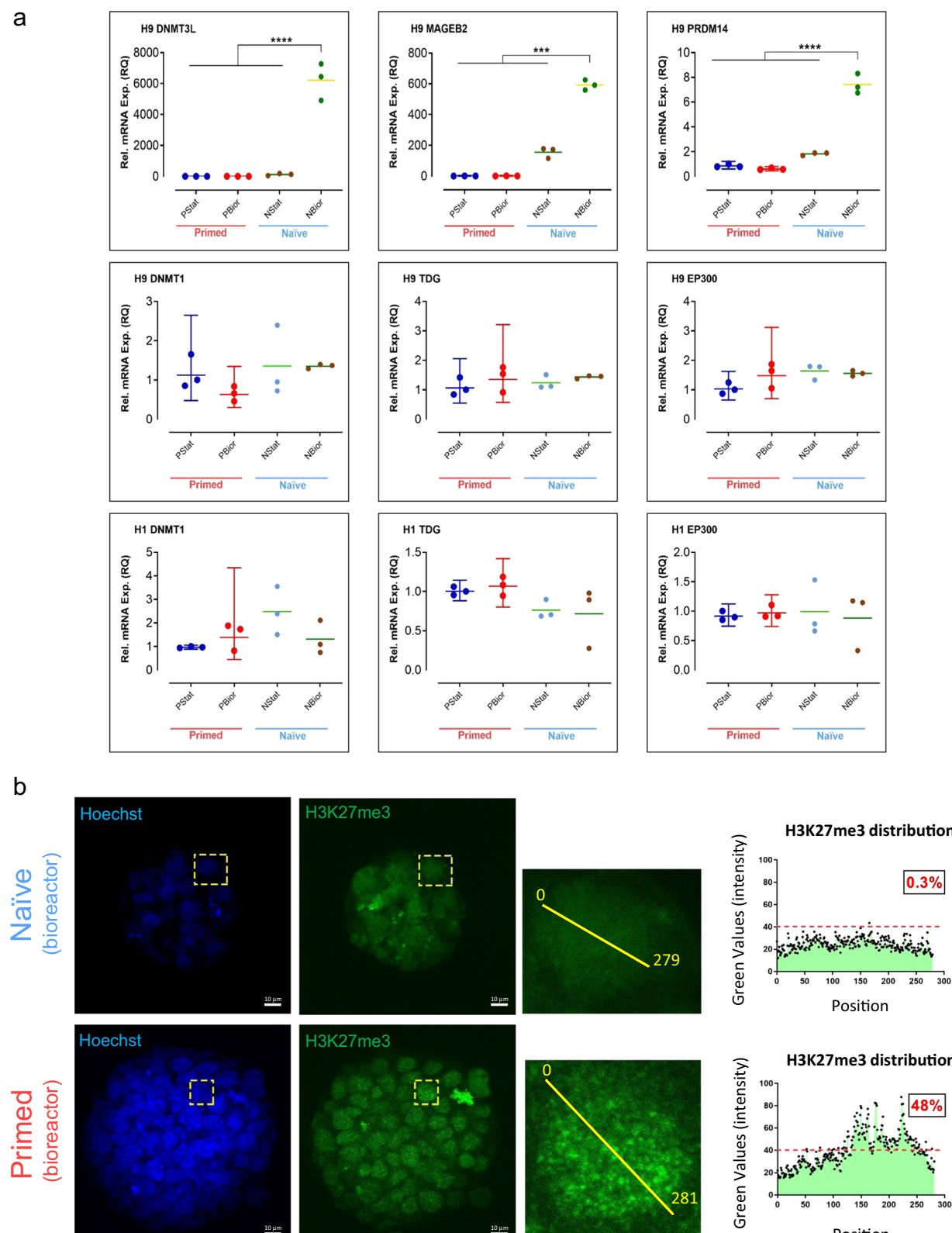

on static condition compared to their original statically cultured cells (Fig. 10c). A single event of structural chromosomal aberration (only in one metaphase spread) was shown in the original statically cultured, primed hPSCs, while there was no clonal occurrence of structural chromosomal aberrations in bioreactor-cultured, naïve and primed hPSCs following recovery in static culture (Fig. 10c).

## Discussion

The primary goal of this project was to investigate whether converting cell-state from primed to naïve pluripotency influences bioprocessing and whether this can resolve current hPSC manufacturing gaps.

Converted naïve hPSCs showed significantly greater growth rates and multiplication ratios as well as shorter doubling times

**Fig. 5 Epigenetic characterization of naïve and primed hPSCs. a** Expression of epigenetic regulators in naïve and primed hPSCs. Depiction of DNMT3L, MAGEB2, PRDM14, DNMT1, TDG, and EP300 epigenetic regulator transcript levels for H1 and H9 hESC lines analyzed by RT-qPCR. The aliquots of statically cultured, cells for each naïve (P4) and primed hPSC sample were collected for gene expression analysis before inoculating them into stirred suspension bioreactors. Those cells were counted as "NStat" and "PStat". The aggregates of day 4 post-inoculation were collected for each naïve and primed hPSC sample for gene expression analysis. Expression was quantified relative to the housekeeping gene GAPDH and was normalized to statically cultured, primed hPSC (PStat) level (=1). All the cultures underwent fed-batch condition (60% media change, 48-h post-inoculation) for bioreactor cultures. The data presented are generated from inoculation density of 50,000 cells/mL. The data are represented as mean ± SEM ($n = 3$). **** = Adj-P < 0.0001, *** = Adj-P = 0.0002 and 0.0004, ** = Adj-P = 0.0021, * = Adj-P = 0.0291 using GraphPad Prism. Adj-P = adjusted $p$ value. RQ = relative quantification. PStat = primed hPSCs cultured under static condition. PBior = primed hPSCs cultured under stirred suspension bioreactor condition. NStat = naïve hPSCs cultured under static condition. NBior = naïve hPSCs cultured under stirred suspension bioreactor condition. **b** Intensity and distribution of H3K27me3 foci in the nuclei of bioreactor-cultured, naïve and primed H9 hPSC aggregates. Representative confocal images were obtained after whole-mount immunofluorescence for H3K27me3 on day 4 post-inoculated naïve and primed hPSC aggregates. Alexa Fluor 488 was used as secondary antibody. Scale bars = 10 μm. Intensity and distribution of H3K27me3 foci were analyzed by Image J using Plot Profile analysis. Enlarged insets from selected nuclei per naïve and primed hPSC aggregates are shown on the right side. A symmetric midline was applied in naïve and primed hPSC aggregate's nuclei (within the enlarged images) using Plot Profile analysis in Image J to measure H3K27me3 intensity and distribution foci along the indicated midline. 279 foci for naïve hPSC nuclei and 281 foci for primed hPSC nuclei were analyzed along the midline. The H3K27me3 distribution graphs show the intensities of foci along the midline within the enlarged images. The x-axis represents the position of analyzed foci along the midline and the y-axis reflects the intensity value of the analyzed foci along the midline. The indicated numbers (%) within the black boxes show the percentage of dots that are located above the indicated arbitrary red line (the percentage of dots located above 40).

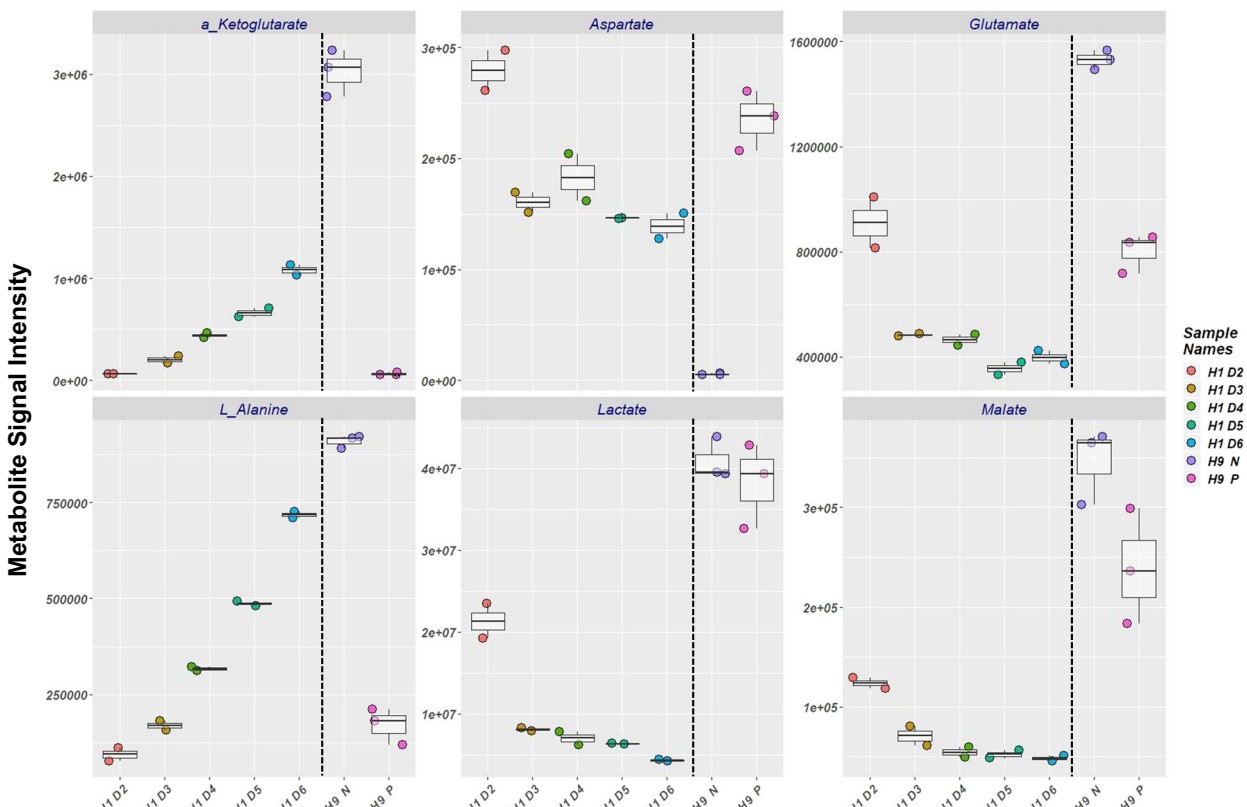

**Fig. 6 Metabolomics of naïve and primed hPSCs in stirred suspension bioreactor.** Depiction of metabolite intensity in the media of naïve and primed hPSCs cultured in bioreactor analyzed by high-resolution LC-MS semi-targeted metabolomics. Extracellular metabolites in the media of H1 hESC line were tracked from day 2 (H1 D2) to day 6 (H1 D6) of bioreactor culture as the media was changed gradually from mTESR1 to RSeT from day 2 to day 6. The media was collected every day for LC-MS analysis. Extracellular metabolites in the media of established naïve (P5, already converted and passaged several times on static condition before inoculation) (H9 N) and primed (H9 P) hPSCs from H9 hESC line were tracked at day 4 of bioreactor culture (the last two columns of the graph). The data presented are generated from an inoculation density of 50,000 cells/mL.

compared to their primed counterparts. Following inoculation into stirred suspension bioreactors, naïve hPSCs recovered even under unoptimized batch culture conditions. Furthermore, primed hPSCs exhibited poor aggregation with many cells remaining single, compared to naïve hPSCs. These results demonstrate that compared to conventional hPSCs, naïve hPSCs form aggregates more efficiently than their primed complement

and show a greater capacity for aggregate growth in stirred suspension culture.

Our several attempts to serial passage naïve hPSCs in different shaken suspension systems such as 6-well plates and 100 mL stirred suspension bioreactors showed little success. In these systems, following the first few days of passage two, the aggregates began to fall apart. Since we were able to recover the naïve cells on

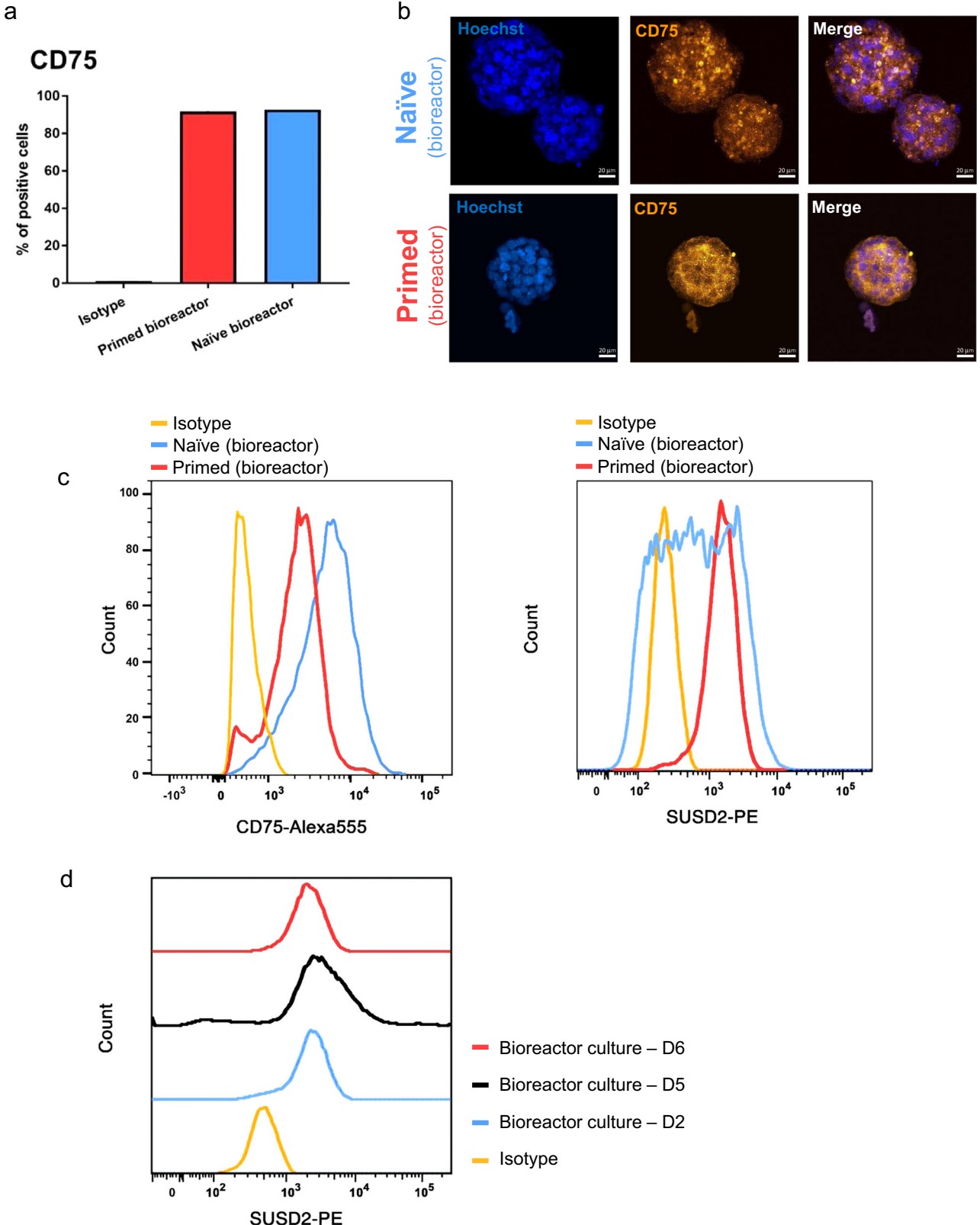

static MEFs following bioreactor culture (Supplementary Fig. 2g), we believe that it is the extended exposure to suspension culture that affects the ability of cells to proliferate, rather than the end of the bioreactor passage procedure. Others have also been unsuccessful in maintaining pluripotency when passaging naïve hPSCs from bioreactor to bioreactor[26]. Lipsitz et al. carried out an

in-depth media investigation and alteration to successfully serial passage alternative hPSCs in suspension well-plates, eventually removing the ERK inhibitory cytokine PD[26]. In fact, the serial passaging of alternative hPSCs in their suspension system was dependent on the ERK inhibitory cytokine's withdrawal from suspension culture. We believe that converting cell state from

**Fig. 7 Expression of naïve pluripotency cell-surface protein markers in bioreactor-cultured, naïve and primed hPSCs. a** Average percentages of CD75-positive cells in bioreactor-cultured, naïve and primed H9 hPSCs. The data are represented as mean percentage of positive cells ($n = 2$). **b** Representative immunofluorescent images showing CD75 expression in bioreactor-cultured, naïve and primed H9 hPSC aggregates. Alexa Fluor 555 was used as the secondary antibody. Scale bars = 20 μm. **c** Histogram of flow cytometry analysis showing fluorescently labeled CD75, and SUSD2 in bioreactor-cultured, naïve and primed H9 hPSCs. Gates were set based on an isotype control. Aggregates of day 4 post-inoculation were used for flow cytometry analysis and for confocal imaging for both naïve and primed hPSCs. All the cultures underwent fed-batch condition (60% media change, 48-h post-inoculation) for bioreactor cultures. The data presented are generated from an inoculation density of 50,000 cells/mL. **d** Expression dynamics of SUSD2 cell-surface protein during primed to naïve pluripotency conversion in the stirred suspension bioreactor within six days of culture. The media was changed gradually from mTESR1 to RSeT from day 2 to day 6 of culture in the bioreactor, and the cells were collected every day for analysis.

primed to naïve pluripotency enhances hPSC bioprocessing properties; however, the success of long-term, serial passaging of naïve hPSC in bioreactors will require the development of specialized media. The RSeT naïve hPSC media used throughout these experiments was not sufficient for long-term bioreactor culturing and requires component optimization, similar to how STEMCELL Technologies Inc. produced an mTeSR™3D media for suspension culture of primed hPSCs.

Gene expression analysis demonstrated elevated naïve- and diminished primed- pluripotency-associated transcripts in our bioreactor-cultured, naïve hPSCs. We performed an extensive transcriptomic analysis of naïve hPSCs in suspension by incorporating both static suspension and stirred suspension culture into the RNA-seq study. Transcriptomic results comparing static, static suspension, and stirred suspension culture showed that the static suspension could not maintain naïve pluripotency, suggesting that cell–cell connections alone are not sufficient to maintain naïve pluripotency. Exclusive enrichment of the mechano-sensing HIPPO signalling in the stirred suspension culture suggests the importance of a mechanical environment for maintaining naïve pluripotency in the bioreactor. HIPPO signalling is regulated by cell–cell contact and mechanical force[56]. Its exclusive enrichment in stirred suspension culture implies that three-dimensional cell–cell connections combined with fluid shear stress may be required to maintain the naïve pluripotent state. Compared to the static condition, both static suspension and stirred suspension culture displayed downregulation of pathways related to cell–cell connection, adhesion, and ECM. This supports the hypothesis that common differentially expressed genes in suspension culture are not the specific contributing factors in maintaining naïve pluripotency in stirred suspension bioreactor. Reduced expression of adhesion-related genes in suspension culture (due to the cell's lower dependence on extracellular adhesion for survival) has been already shown[11]. However, the loss of pluripotency that we observed in static suspension suggests that cell–cell connections alone are not sufficient to maintain naïve pluripotency. Higher enrichment of terms related to cellular response to stress and intracellular movement in stirred suspension bioreactor support the requirement of fluid shear stress to maintain the naïve pluripotent state. These all support the positive influence of the bioreactor environment on naïve pluripotency and that the naïve pluripotency state is maintained in response to mechanical forces in stirred suspension culture.

Naïve hPSCs also revealed the expression of epigenetic regulatory transcripts associated with naïve pluripotency in stirred suspension bioreactors. Bioreactor-cultured, naïve hPSCs showed elevated expression of *DNTMT3L*, *MAGEB2*, *PRDM14*, and *PTEN* signaling (in our RNA-seq data). In naïve hPSCs, these regulators are mostly involved in DNA demethylation and primordial germ cell (PGC) specification[57–59], two functions which are hallmarks of naïve pluripotency and have been observed before[44,45,54,60–62]. High expression levels of these regulatory genes suggest that the bioreactor may support naïve pluripotency maintenance.

Moreover, naïve hPSC aggregates exhibited hallmarks of X-chromosome reactivation (less condensed H3K27me3 foci) in the bioreactor. However, the epigenetic regulators *DNMT1*, *TDG*, and *EP300* demonstrated no difference between bioreactor-cultured, naïve hPSCs and their primed equivalents. Albeit, there was a discrepancy in the expression level of the de novo DNA methyltransferase *DNMT3B* in H1 and H9 naïve hPSCs in the bioreactor. There is controversial data surrounding the expression levels of these epigenetic regulators according to various naïve pluripotency media compositions[20,21,35,36]. For instance, according to different naïve pluripotency conditions (e.g., NHSM, RSeT, T2iLGO, and NJSM) and stages, the expression levels of *DNMT3L* and *DNMT1* were shown to be different[20,35,36,54]. RSeT media used in this study has previously been shown to promote naïve pluripotency features with intermediate expression of epigenetic regulatory transcripts (e.g., *DNMT3L* and *DNMT3B*)[35]. Considering this, our data may support the hypothesis that the bioreactor helps maintain naïve pluripotency. However, there may be heterogeneous populations of naïve cells harboring different epigenetic and naïve pluripotency stages in the bioreactor. This is also in line with our flow cytometry data that showed wide distributions of naïve and primed surface markers in bioreactor-cultured, naïve hPSCs. One possibility is that the cells at the periphery of naïve hPSC aggregates are exposed to the greatest fluid shear stress in the bioreactor and might show different expression levels of naïve pluripotency (similar to nuclear localization of TFE3 in the outer layer of bioreactor-cultured, primed hPSC aggregates). RSeT itself might be involved in generating heterogeneous or intermediate naïve pluripotency in the bioreactor. However, truly exploring such possibility and the entity of those heterogenous cells requires a more detailed investigation with extensive high-throughput analyses, such as single-cell (sc) RNA-seq (scRNA-seq)[24] and Methyl-Seq target enrichment bisulfite sequencing for differentially methylated regions[36]. A deeper understanding of the specific epigenetic state of naïve hPSCs in the bioreactor may be beneficial for long-term expansion. For example, specific epigenetic modulators that maintain an open chromatin state could be incorporated.

Dynamics of metabolites revealed robust production of the key naïve pluripotency metabolites α-KG and glutamate[45,46,62] in converting and established naïve hPSCs. α-KG has been reported to promote histone/DNA demethylation and promotes the maintenance of naïve pluripotency[45,46,62]. In line with previous studies[45], aspartate levels dropped significantly in spent bioreactor media of established naïve hPSCs. In contrast, lactate and malate remained comparable in the media of both established naïve and primed hPSCs. Yet, malate showed a trend of increasing in the media of established naïve hPSCs. The main metabolite of naïve pluripotency, α-KG, exhibited robust production in converting and established naïve hPSCs while the dynamics of aspartate, glutamate, lactate, and malate were all different in converting and established naïve hPSCs. The changing dynamics of these metabolites (e.g., glutamate and malate) could reflect the difference in media change between gradually

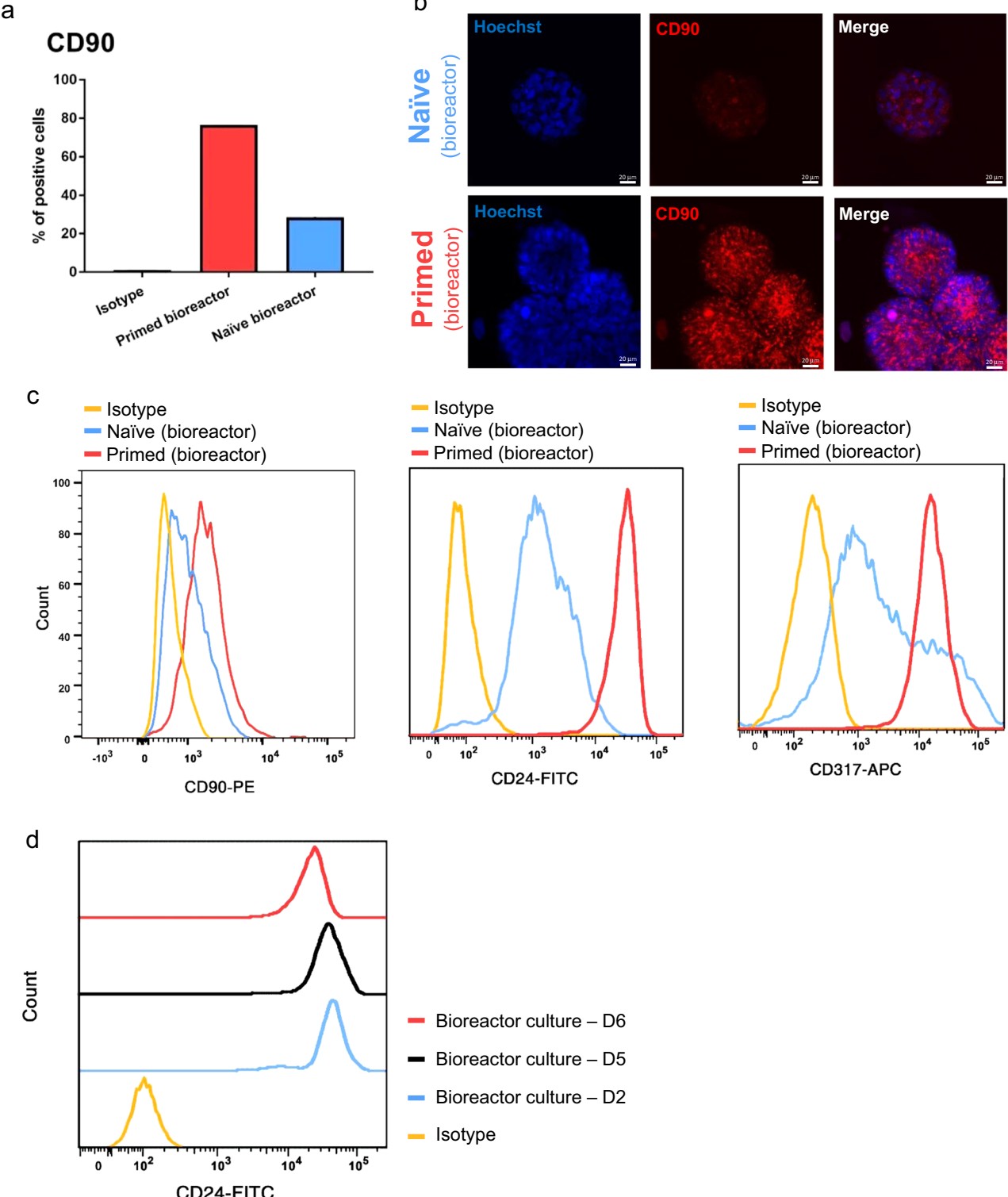

**Fig. 8 Expression of primed pluripotency cell-surface protein markers in bioreactor-cultured, naïve and primed hPSCs. a** Average percentages of CD90-positive cells in bioreactor-cultured, naïve and primed H9 hPSCs. The data are represented as mean percentage of positive cells (*n* = 2). **b** Representative immunofluorescent images showing CD90 expression in bioreactor-cultured, naïve and primed H9 hPSC aggregates. CD90 antibody was conjugated with R-phycoerythrin (R-PE). Scale bars = 20 μm. **c** Histogram of flow cytometry analysis showing fluorescently labeled CD90, CD24, and CD317 in bioreactor-cultured, naïve and primed H9 hPSCs. Gates were drawn based on an isotype control. Aggregates of day 4 post-inoculation were used for flow cytometry analysis and for confocal imaging for both naïve and primed hPSCs. All the cultures underwent fed-batch condition (60% media change, 48-h post-inoculation) for bioreactor cultures. The data presented are generated from inoculation density of 50,000 cells/mL. **d** Expression dynamics of CD24 cell-surface protein during primed to naïve pluripotency conversion in the stirred suspension bioreactor within six days of culture. The media was changed gradually from mTESR1 to RSeT from day 2 to day 6 of culture in the bioreactor, and the cells were collected every day for analysis.

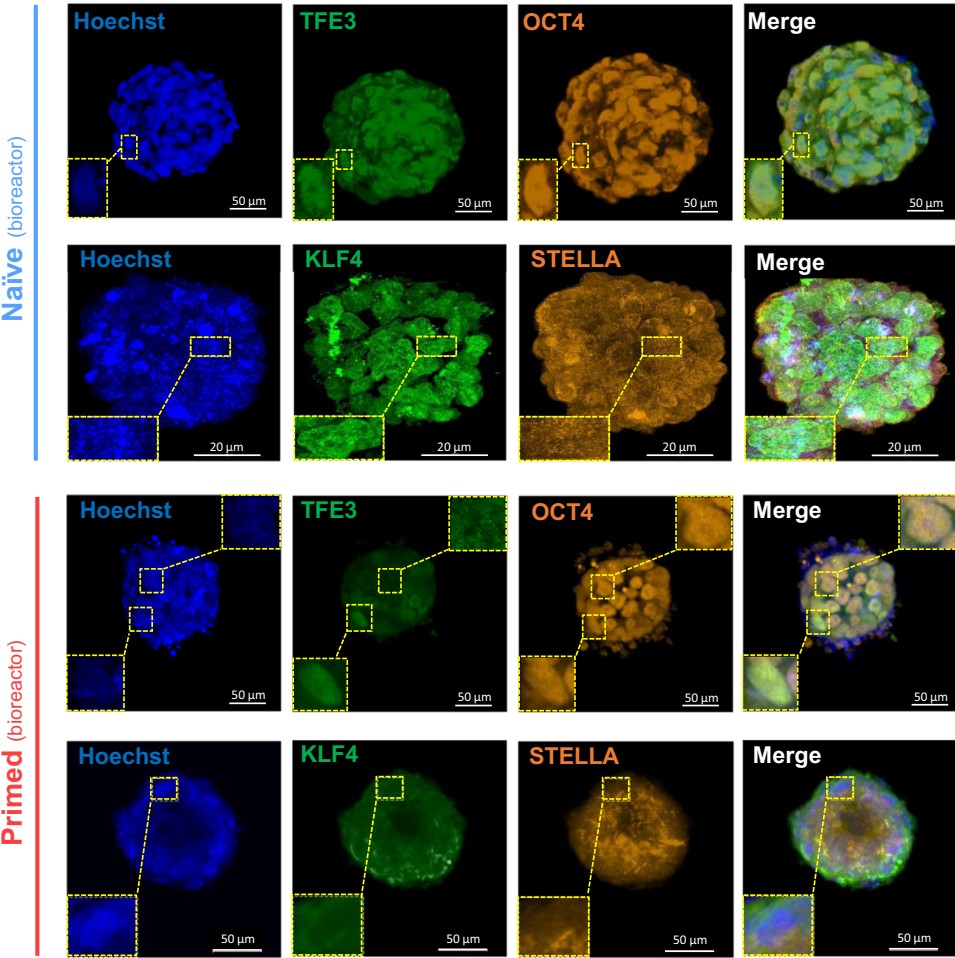

**Fig. 9 Whole-mount immunostaining of bioreactor-cultured, naïve and primed hPSC aggregates for pluripotency and naïve-related markers.**
Representative confocal images are shown for naïve and primed H9 hPSC aggregates double-immuno-stained for TFE3/OCT4 and KLF4/STELLA. Insets are enlargements of the dashed boxes. The aggregates on day 4 post-inoculation were used for whole-mount staining and confocal imaging for both naïve and primed hPSC sample. For naïve hPSC aggregates immuno-stained with KLF4/STELLA, scale bars = 20 μm. For all other images, scale bars = 50 μm.

converting and established naïve hPSCs in the bioreactor. The media for H1 was gradually changed from mTeSR1 to RSeT (primed-to-naïve) from day 2 to 6 of culture in bioreactor, while in established naïve (P5), the media used every day in the bioreactors was RSeT (naïve pluripotency media). This media change might affect the homeostasis of glutamate and malate, (e.g., glucose/glutamine-dependent glutamate and malate production), leading to different patterns in converting and established naïve hPSCs[45,46,63,64]. Furthermore, cell-state status in converting and established naïve hPSCs in the bioreactor could affect metabolite dynamics. The established naïve (H9 N) hPSCs were already established in static culture using RSeT media (4–5 passages) before inoculating into the bioreactor. While for H1, the media gradually changed in the bioreactor from primed-to-naïve pluripotency media to convert them toward a naïve state of pluripotency. Although, we have fully established naïve hPSCs for H9, we believe there might be a heterogeneous population of naïve-primed hPSCs in gradually converting H1 condition (as they were not fully established naïve hPSCs). Given that naïve and primed hPSCs have distinct metabolic dynamics (oxidative phosphorylation vs. glycolysis)[65] and growth kinetics[64,66] (Fig. 2), this potential heterogeneous naïve-primed population might affect the concentration of glutamate and malate in converting and established naïve hPSCs. Alternatively, the media shift in the bioreactor in converting naïve hPSCs may have caused extracellular stress. In this scenario, cellular adaptation in response to the

newly changed media might affect cell-state condition (e.g., cell cycle) and contribute to a different pattern of glutamate and malate[64,67,68].

Taken together, our metabolomics data indicate that the bioreactor environment alongside a specific media (RSeT in our case) helps to maintain a naïve pluripotent state in hPSCs. Optimizing metabolite level and feeding regime, developing specific 3D naïve pluripotency media, and performing additional metabolomic bioprocessing can further improve future biomanufacturing of naïve hPSCs.

The discrimination of naïve and primed pluripotency cell-surface markers has been controversial, and their expression pattern can be different according to the various naïve pluripotency protocols[36,47]. We tried to choose markers from recent reports[25,49] to characterize naïve and primed hPSCs in stirred suspension culture. Our flow cytometry analysis did not show a remarkable distinction of the naïve pluripotency surface markers CD75 and SUSD2 between bioreactor-cultured, naïve and primed hPSCs. Yet, naïve pluripotency surface markers revealed wider distribution in bioreactor-cultured, naïve hPSCs. Expression dynamic of SUSD2 also remained stable with no significant shift within six days of bioreactor culture. However, all of the primed pluripotency cell-surface markers CD90, CD24, and CD317 revealed significant downregulation in naïve hPSCs in the bioreactor. Of note, the expression dynamic of CD24 showed downregulation on day 6 of bioreactor culture (persisted until

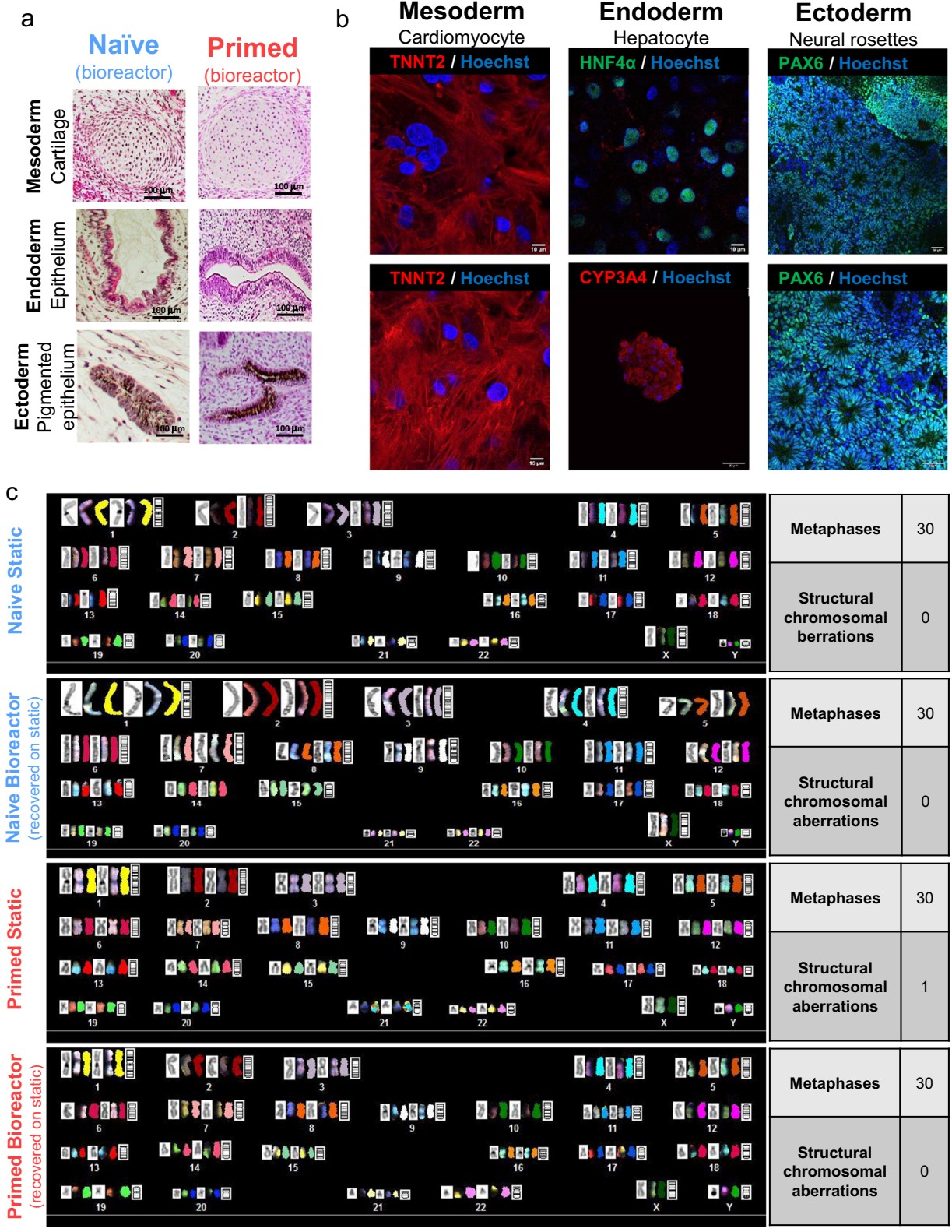

day 5). This is in line with our transcriptomic data, which revealed a downregulation of primed pluripotency biological pathways. These all suggest that the bioreactor environment helps maintain naïve pluripotency. More specifically, reduced primed pluripotency hallmarks (transcripts, biological pathways, specific cell-surface markers) are supportive of naïve pluripotency maintenance in the bioreactor. Previous studies on naïve pluripotency surface markers demonstrated that the cells cultured in

RSeT media represented an intermediate naïve human pluripotency state[35,36,47], displaying downregulation of primed pluripotency surface markers (e.g., CD24 and CD90) but no upregulation of naïve pluripotency surface markers (e.g., CD75)[47] (similar to our results). Given that RSeT media generates intermediate naïve hPSCs[35,36,47], we believe this could be one of the reasons there is no difference in the expression of naïve pluripotency cell-surface markers in the bioreactor. Additionally, the

**Fig. 10 Teratoma formation, targeted in vitro differentiation, and chromosomal stability of bioreactor-cultured, naïve and primed hPSCs. a** Teratomas generated after the injection of naïve and primed H9 hPSCs subcutaneously into the hind legs of CB17 SCID mice. Representative images for each germ layer are shown. Scale bars = 100 µm. **b** Representative confocal images are shown for cardiomyocytes (immuno-stained for TNNT2), hepatocytes (immuno-stained for HNF4α), and neural rosettes (immuno-stained for PAX6) differentiated from bioreactor-cultured, naïve H1 hPSCs. Scale bars = 10 µm and 50 µm. **c** Representative karyotypes of bioreactor-cultured, naïve and primed H1 hPSCs following recovery on static condition compared to their original statically cultured cells, analyzed via spectral karyotyping. Tables summarizing the karyotype analyses (SKY) for structural chromosomal aberrations in naïve and primed hPSCs are shown. The aggregates of day 4 post-inoculation were used for teratoma formation, targeted in vitro differentiation, and karyotype analysis for both naïve and primed hPSC samples. All the cultures underwent fed-batch condition (60% media change, 48-h post-inoculation) for bioreactor cultures. The data presented are generated from an inoculation density of 50,000 cells/mL.

wider distribution of our studied naïve and primed pluripotency cell-surface markers in bioreactor-cultured, naïve hPSCs might suggest the possibility of a heterogeneous population of naïve cells in the bioreactor. This could account for our findings regarding naïve hPSC surface markers in the bioreactor. Future studies are warranted to explore this possibility.

Comparable to previously reported work on naïve pluripotency[20,50], bioreactor-cultured, naïve hPSC aggregates demonstrated nuclear expression and localization of TFE3 and KLF4, and revealed efficiently targeted in vitro differentiation into cardiomyocytes, hepatocytes and neural rosettes. Interestingly, bioreactor-cultured, naïve hPSCs displayed more rapid teratoma formation than their primed counterparts, comparable to the kinetics of mouse PSC teratoma formation following bioreactor culture[8,69]. Similar to our previous bioreactor inoculation data[8,69] showing that naïve mouse PSCs do not display a lag phase following inoculation, this enhancement in teratoma formation kinetics suggests that such a lag phase also does not occur upon transplantation of naïve hPSCs. Bioreactor-cultured, naïve hPSCs also showed other previously reported hallmarks of naïve pluripotency[29,51], such as a higher propensity for differentiation into the endoderm lineage and a reduced capacity for mesoderm lineage differentiation. Furthermore, our results showed no occurrence of structural chromosomal aberrations in bioreactor-cultured, naïve hPSCs post-recovery in static condition. More detailed investigations are essential to effectively analyze the genomic stability of naïve and primed hPSCs.

The apparent heterogeneity of our naïve hPSC population in stirred suspension bioreactors, specifically based on the discrepancy in the expression of epigenetic regulators and cell-specific surface markers, raises the question as to whether these cells reside in an alternative naïve-like pluripotent state rather than a fully naïve one. Defining exact criteria for "full naïve" pluripotency of hPSCs is an area of active research in the field[19] . An optimal standard condition for naïve pluripotency in static culture has remained elusive[19], and the field is even newer for dynamic suspension bioreactors. As mentioned, assigning this status based on cell-surface markers, and epigenetic regulator expression has been controversial, and while our metabolomics analysis did show changes during the conversion of H1 cells, some metabolites were not consistent between statically established H9 naïve cells and our converting H1 cells in the bioreactor. Furthermore, we used a commercial media (RSeT) from STEMCELL Technologies Inc. that has been shown extensively to be sufficient for reversion of primed to an intermediate naïve-like state [35,36,47].

As others have done[24,25,43,46,47], we used transcriptomic, expression of epigenetic regulator, metabolomic, and specific cell-state surface marker analyses to define naïve pluripotency. Our results do support the case that our naïve hPSCs in bioreactor recapitulate many features of naïve pluripotency, showing distinct features compared to their primed counterparts, indicative of a naïve pluripotent state. While we cannot say for certain that our cells are "fully" naïve, we are confident they represent a naïve-like state with properties amenable to scalable bioreactor production.

In summary, our data corroborate that the conversion of hPSCs from a primed towards a naïve pluripotent state enhances hPSC manufacturing in suspension culture. This provides opportunities to pursue cell-state conversion as a potential strategy to overcome limitations in manufacturing of hPSCs for cell therapy approaches. Additionally, via transcriptomic, epigenetic regulators expression, metabolomics, and cell-surface protein marker expression analyses, we demonstrate that the bioreactor's mechanical environment helps maintain naïve pluripotency while suspension culture alone does not. This positive influence of the bioreactor environment on maintaining naïve pluripotency suggests it may be feasible to convert hPSCs to the naïve pluripotent state in the bioreactor. Furthermore, we are excited about the prospect of expanding naïve hPSCs in the bioreactor using fewer cytokines, as we have done previously using mPSCs[70]. Optimizations such as these can improve hPSC manufacturing and make these cells a more viable option for the clinic.

Future experiments should look at optimizing traditional bioprocess variables, including bioreactor geometry, inoculation conditions, agitation rates, feeding regimes, growth factor consumption, pH level, and dissolved oxygen concentrations. In particular, we propose focusing on scaling-up production to computer-controlled, on-line bioreactor systems, which can be used to control the physicochemical environment (pH and dissolved gas), nutrient and metabolite concentrations, and growth factors[71]. Computer-controlled bioreactors will be imperative for translation to clinical applications and for establishing Good Manufacturing Process (GMP)-compatible bioprocesses[72].

## Methods

**Culturing of primed human embryonic stem cells (hESCs).** H1 and H9 (WiCell) hESCs were maintained in standard culture conditions (37 °C, 5% $CO_2$) on Corning® Matrigel® hESC qualified Matrix (BD Biosciences, BD354277)-coated 60-mm tissue culture dishes with mTeSR™1 medium per the manufacturer's recommendations (Stem Cell Technologies, 85875). Primed hESC cultures were fed every day with fresh mTeSR™1 medium, and the confluent cell cultures (usually 6–7 days) were passaged either as small clumps via Dispase (1 U/mL) (Stem Cell Technologies, 7923) or as single cells via StemPro® Accutase® Cell Dissociation Reagent (Thermo Fisher Scientific, A1110501) and were split at a ratio of 1:6. We have complied with all relevant ethical regulations and obtained approval from human ethics protocol and Canadian Institutes of Health Research's stem cell oversight committee.

**Conversion of primed hESCs to a naïve state.** The primed hESC lines (H1 and H9) were converted to naïve pluripotency using RSeT™ Medium according to manufacturer's instructions (Stem Cell Technologies, 5970). Briefly, once the primed hESCs reached to 70–80% confluency, they were passaged as small clumps (approximately 50–200 µm in size) using Dispase (1 U/mL) (Stem Cell Technologies, 7923), and were transferred onto mitomycin C treated-mitotically inactivated MEFs (iMEFs)-coated plates containing mTeSR™1 medium. The cells were incubated at 37 °C for 24–36 h under normoxic conditions (20% $O_2$, 5% CO2) in mTeSR™1 medium. The mTeSR™1 medium was then replaced with RSeT™ medium and a daily full medium change with RSeT™ was done during the reversion to naïve hPSCs. Upon the appearance of domed-shaped and tightly packed colonies (after 3–4 days in RSeT™ Medium), the colonies were typically ready to be passaged. The culture was passaged using StemPro® Accutase® Cell Dissociation Reagent (Thermo Fisher Scientific, A1110501), and seeded on new iMEFs containing RSeT™ medium supplemented with Y-27632 (Stem Cell Technologies, 72304) to a final

concentration of 10 μM. During the first passages and reversion in RSeT™ condition, the cells were split at a ratio of 1:6. Once the culture was established, cells were split at a ratio of 1:10 every three days. The newly formed (P0 from static culture) and established (P4 from static culture) naïve hPSCs were collected and utilized for inoculation into the stirred suspension bioreactors and suspension well-plates as well as other downstream analyses.

**Cell inoculation into stirred suspension bioreactors and static suspension well-plates.** Naïve and primed hESCs from static culture were inoculated into 100 mL stirred suspension bioreactors (NDS Technologies, Vinland, NJ, USA) at 5E4 cells/mL. To elucidate the effect of various inoculation cell densities, naïve hPSCs were inoculated into 100 mL stirred suspension bioreactors at 1E4, 2.5E4, and 5E4 cells/mL. Naïve hPSCs were cultured in 24 and 6-well shaken suspension plates at seeding densities of 1E5 and 2E5 cells/mL, respectively, to examine higher suspension seeding densities. For serial passaging, naïve hPSCs were inoculated into 100 mL stirred suspension bioreactors at 5E4 cells/mL, and at 1E5 cells/mL and 2E5 cells/mL into 24 and 6-well shaken suspension plates, respectively.

Confluent primed hESCs from the static culture were enzymatically dissociated into single cells using StemPro® Accutase® Cell Dissociation Reagent and mechanical pipetting. Four 50 μL samples were then taken for cell counts using a trypan blue exclusion assay. The desired cell concentration was then added to 6-well static suspension culture dishes (Greiner Bio-One, 657185) containing 2 mL of complete mTeSR™1 medium supplemented with 10 μM Y-27632. Following 1 h of static suspension, small cell clumps were observed in the culture wells. These clumps were gently pipetted up and down several times and added to the 100 mL bioreactors containing mTeSR™1 medium supplemented with 10 μM Y-27632 and 0.1 nM Rapamycin (Sigma–Aldrich, R8781) at an agitation rate of 100 rpm. A similar procedure was used for the bioreactor inoculation of naïve hESCs, though an additional step was required for iMEF removal. Following dissociation into single cells, the naïve hESC and iMEF mixture were seeded onto gelatin (Sigma–Aldrich, G1890) coated dishes with complete RSeT™ medium for a period of 45 min–1 h. During this time, the iMEFs would adhere to the gelatin coated dishes while the naïve hESCs remained suspended. The naïve hESC suspension was aspirated from the dishes and centrifuged at 200 × g for 5 min. Like the primed hESC culture, naïve hESCs were counted and seeded into suspension wells supplemented with Y-27632 for a period of 1 h before being added to the 100 mL stirred suspension bioreactors. Bioreactor naïve hESC culture medium was RSeT™ medium supplemented with 10 μM Y-27632 and 0.1 nM Rapamycin. The agitation rate of 100 rpm was used for hPSCs cultured in 100 mL stirred suspension bioreactor, and an agitation rate of 100 rpm (orbital shaker) was used for naïve hPSCs cultured in 24 and 6-well shaken suspension plates.

Naïve and primed hESCs were cultured in 100 mL bioreactors for a period of five days under both batch and fed-batch conditions. For 24 and 6-well shaken suspension plates, naïve hPSCs were cultured for five days under fed-batch condition. The medium change was not performed for the cells cultured under batch conditions. The cells cultured in fed-batch conditions underwent a 60% medium change (only containing Rapamycin and no additional Y-27632) on day 2.

**Analysis of cell growth kinetics.** To analyze growth kinetics of cells in stirred suspension bioreactor and shaken suspension well-plates, cell counts were performed each day (D0–D5) post-inoculation. Aggregate collection was done at day 1–5 post-inoculation. For 100 mL stirred suspension bioreactors, two 5.0 mL samples were obtained from each bioreactor (n = 4 per group, per time point; day 1–5 post-inoculation), and then centrifuged. After centrifugation, the pellet was resuspended and incubated with 1.0 mL of StemPro® Accutase® Cell Dissociation Reagent at 37 °C for 5 min to dissociate the aggregates. Medium was then added to neutralize the reaction and cells were centrifuged and resuspended for counts. Cell viabilities and densities were determined using a hemocytometer and a standard trypan blue exclusion test. Each sample was dissociated and counted twice. For all cells cultured in shaken suspension well-plates, two wells were harvested from each condition each day for counts. Medium and cells were aspirated from each well into respective conical tubes. Each well was then rinsed twice with medium that was added to the respective conical tube. Aggregate dissociation and viability cell counts were performed as described above. The results were used to generate viable cell density growth curves. The multiplication ratio was defined as the final viable cell density divided by the inoculation density. The exponential doubling time and the exponential growth rate over the course of the passage were determined using Eqs. (1) and (2):

$$X_2 = X_1 e^{-\mu \Delta t} \tag{1}$$

$$t_D = \ln(2)/\mu \tag{2}$$

where $X$ is the viable cell density at a given time point (cells/mL), $\mu$ is the growth rate (h$^{-1}$), $t$ is the time in hours, and $td$ is the population doubling time (h).

**Assessment of average aggregate size and distribution.** Each day (D1–D5 post-inoculation), 1.0 mL samples were taken from stirred suspension bioreactors (naïve and primed hPSC cultures) to be imaged with a Ziess Axiovert 25 microscope (Carl Zeiss AG, Oberkochen, Germany). Average aggregate diameters were

determined by measuring 60 aggregates from the microscope images using Axio-Vision Rel. 4.8. Day 5 aggregate size distributions were measured using the Beckman Coulter Multisizer III (Miami, FL, USA) with a 1000 μm aperture tube, which has an effective particle-size range of 20–600 μm. Aggregate samples collected from the bioreactors were resuspended in a solution containing an 80:20 (v/v) mixture of Isoton II (Beckman Coulter) and glycerol (Fisher Scientific). The amplitude of the voltage pulse associated with each particle was converted to a particle size based on a calibration coefficient using L90 Latex Particle Standard beads (Beckman Coulter) with a nominal size of 90 μm, with the bin size set to 128, and bin spacing on a log diameter. Following collection, data were analyzed to assess aggregate size distributions.

**RNA isolation, reverse transcription (RT) quantitative (q) polymerase chain reaction (PCR) (RT-qPCR), and RT-PCR.** Naïve and primed hPSCs from both static and stirred suspension (on day 4 post-inoculation) cultures were collected and used for RNA isolation. Total RNA was extracted using PureLink™ RNA Mini Kit (Thermo Fisher Scientific, 12183018A) according to the manufacturers' protocol, followed by DNAse I digestion using DNAse I Amplification Grade (Thermo Fisher Scientific, 18068015), and then 500 ng RNA was used for cDNA synthesis using Superscript® IV Reverse Transcriptase (Thermo Fisher Scientific, 18090010) and Oligo(dT)20 Primer (50 μM) (Thermo Fisher Scientific, 18418020) according to the manufacturers' instructions. To quantitate transcripts, the subsequent RT-qPCR gene expression analysis was performed on Applied Biosystems (Thermo Fisher Scientific) either Fast SYBR™ Green Master Mix (Thermo Fisher Scientific, 4385612) or TaqMan™ Universal PCR Master Mix, no AmpErase™ UNG (Thermo Fisher Scientific, 4324018). For each sample, relative mRNA expression was quantified relative to the housekeeping gene GAPDH and was normalized to statically cultured, primed hPSC level (=1). The relative quantification (RQ) was done based on comparative $C_T$ ($\Delta\Delta C_T$) through $2^{-\Delta\Delta CT}$ method. The gene expression results were shown as relative mRNA expression (RQ) to statically cultured, primed hPSCs (RQ = 1). At least three technical and biological replicates were assayed for all quantitative RT-PCR reactions. Naïve pluripotency-associated genes; *KLF2, KLF4, STELLA (DDPA3)*, and *KLF5*, and primed pluripotency-associated genes; *CD24, OTX2, ZIC2*, and *DUSP6*, were used for RT-qPCR. Epigenetic regulator markers used for RT-qPCR were *DNMT3L, MAGEB2, PRDM14, DNMT1, TDG, EP300*, and *DNMT3B*. We used TaqMan probes (TaqMan gene expression assay, Thermo Fisher Scientific) for all markers except *CD24, DNMT1, TDG*, and *EP300*. The TaqMan probes number and primer sequences for SYBR Green probe are listed in Supplementary Data 11. The heatmaps of RT-qPCR were plotted through GraphPad Prism based on the RQ gene expression levels for each marker in each sample normalized to statically cultured, primed hPSC level (=1).

RT-PCR was carried out to evaluate the expression of pluripotency and naïve associated transcripts in naïve hPSCs (passages zero, three, and four after conversion) and primed hPSCs during several passages in static culture. Total RNA was isolated and transcribed into cDNA using the above-mentioned protocol. PCR amplification was performed in a final volume of 20 μl using Taq DNA Polymerase recombinant (Thermo Fisher Scientific, 10342020) and consisted of the following steps: 94 °C for 3 min, 94 °C for 30 s, 55 °C for 45 s and 72 °C for 1 min. Naïve pluripotency-associated markers used for RT-PCR were *XIST* and *KLF4*. Pluripotency related markers used for RT-PCR were *Sox2* and *Nanog*. *GAPDH* was used as a housekeeping gene. Primer sequences used in the amplification reactions are listed in Supplementary Data 11. The PCR products were separated on 1–1.5% agaroses, stained with ethidium bromide, and then visualized and photographed on a UV transilluminator.

**RNA-seq and data analysis.** RNA-seq analysis was done in collaboration with the Centre for Health Genomics and Informatics at the University of Calgary. Naïve hPSCs from static, static suspension and stirred suspension cultures were collected and the total RNA was extracted using PureLink™ RNA Mini Kit (Thermo Fisher Scientific, 12183018A), according to the manufacturers' protocol, including DNAse I digestion using DNAse I Amplification Grade (Thermo Fisher Scientific, 18068015). Concentration and quality of the total extracted RNA was checked using Qubit Fluorometer (QubitR RNA HS high sensitivity assay Thermo Fisher) and TapeStation assay (Agilent 4200 TapeStation System), respectively. The total RNA was depleted of ribosomal RNA using NEBNext® rRNA Depletion Kit (Human/Mouse/Rat) (NEB, E6310L) and 500 ng of the remaining RNA was purified, fragmented, and used for cDNA synthesis and library preparation. The Illumina sequencing library preparation was done using the NEBNext® Ultra™ II Directional RNA Library Prep Kit for Illumina (NEB, E7760L) according to manufacturer's protocol. The constructed libraries were sequenced on an Illumina NextSeq500 generated 75 bp, single-end reads. To process the samples, the quality of raw sequenced reads was checked with FastQC v0.11.5 [73]. No adapter or very highly overrepresented sequences were found, and the quality of sequencing was very high. RNA-seq reads were pseudoaligned to the human Ensembl transcript database; GRCh38 (latest Ensembl human transcript reference) using Kallisto 0.42.4[74]. For the parametric analysis, DESeq (an R/Bioconductor package)[75] was used for differential gene expression using static/static suspension/stirred suspension bioreactor as the explanatory variable. DESeq results were filtered by Adj-p (corrected $p$ value) <0.01. The Ingenuity Pathway Analysis (IPA) z-score algorithm (≥2 or ≤−2, and −log10(Benjamini–Hochberg multiple testing corrected $p$ value of

differential expression) ≥2) was used to identify enriched canonical pathways that are more or less active (positive or negative z-score) according to the IPA database and our RNA-seq data. PCA plots and violin plots of enriched pathways were visualized using R 3.6.3.

**Gene ontology visualization**. Gene ontology analysis was performed using DAVID Bioinformatics Resources v6.8. with significance threshold set at FDR < 0.05. The values for GO visualization were based on –log10($p$ value). The input gene dataset to DAVID was differentially expressed genes from each pairwise condition comparison. GO visualization was performed using Microsoft Excel (version 16.37).

**Metabolomics**. For H1 hESCs, the extracellular metabolite dynamics was analyzed during primed to naïve conversion in the bioreactor. For this, the media of H1 hESCs in the bioreactor was gradually switched from mTESR1 to RSeT media within six days of culture (D2–D6) and the media was collected every day for the analysis. For H9 hESCs, the extracellular metabolites were analyzed in the spent media of primed and established naïve (P5) hPSCs in the bioreactor, and the media of each primed or naïve hPSCs was collected at day 4 of bioreactor culture. The metabolite extraction was started using LC-MS or HPLC grade methanol (Sigma–Aldrich, 1.06035). Briefly, a 950 μL of prechilled 50% MeOH/H2O was added to a 50 μL of the bioreactor-collected media (making a D20 dilution), and the samples were incubated on ice for 30 min to allow for full extraction. Macromolecules were then pelleted by centrifugation at max speed (~18,000–21,000 × $g$) for 10 min in a bench top centrifuge (preferably chilled) to extract the supernatant. Further, the extracted samples were stored in −80 °C prior to running HPLC-MS. For the analysis, 200–400 μL of the extracted supernatant from each sample was transferred into individual wells in 96-well plates and were run in HPLC-MS. General metabolomics runs were performed on a Q Exactive™ Hybrid Quadrupole-Orbitrap™ Mass Spectrometer (Thermo Fisher, IQLAAE-GAAPFALGMAZR) coupled to a Vanquish™ Flex UHPLC System, Integrated biocompatible system (Thermo Fisher, IQLAAAGABHFAPUMBHV). Chromatographic separation was achieved on a Zorbax SB-C18 UHPLC column (2.1 mm × 50 mm × 1.8 um, Agilent, Part number 822700–902) using a binary solvent system at a flow rate of 600 uL/min. Solvent A, 10 mM tributylamine, 10 mM acetate pH 7.5 in 97/3% (v/v) mass spectrometry grade water/methanol; Solvent B, mass spectrometry grade acetonitrile. A sample injection volume of 2 uL was used. The mass spectrometer was run in negative full scan mode at a resolution of 140,000 scanning from 70–1000$m/z$. Data were processed using MAVEN. Visual representation of the data was done in R 3.6.3.

**Flow cytometry**. Naïve and primed hPSCs from both static and stirred suspension cultures were dissociated into single cells using StemPro® Accutase® Cell Dissociation Reagent (Thermo Fisher Scientific, A1110501). To track the dynamics of SUSD2 and CD24 cell-surface markers during primed to naïve conversion, the aggregates were collected from bioreactors every day from day 2 to day 6 of culture. For all other cell-surface marker expression assessment, the aggregates of day 4 post-inoculation were collected. The aggregates were treated with Accutase for 10 min and pipetted to dissociate aggregates into single cells. The cells were washed with cell-specific medium (naïve or primed PSC medium) and were centrifuged at 200 × g for 5 min. The cells were resuspended in 4% paraformaldehyde and were incubated for 15 min at RT. The fixed cells were then washed three times in 4 mL of PBS⁻ (Thermo Fisher Scientific, 14190250) and were blocked in 10% BSA solution (Blocker™ BSA (10X) in PBS⁻, Thermo Fisher Scientific, 37525) at 37 °C for 30 min. The primary antibodies were diluted in blocking solution and were added to the cells and incubated for 60 min at 4 °C. The primary antibodies used for flow cytometry were CD75 antibody (Abcam, ab77676, 1 μg for 1 × 10⁶ cells), SUSD2-PE conjugated antibody (BioLegend, 327406, 5 μl for 1 × 10⁶ cells), CD90-PE conjugated antibody (BD Pharmingen™, 561970, 1 μg for 1 × 10⁶ cells), CD24-FITC conjugated antibody (BioLegend, 311104, 5 μl for 1 × 10⁶ cells), and CD317-APC conjugated antibody (BioLegend, 348410, 5 μl for 1 × 10⁶ cells). After washing the cells with flow buffer containing 0.5% Blocker™ BSA (10X) in PBS⁻, the cells were incubated with secondary antibody Alexa Fluor® 555 (Thermo Fisher Scientific, A-21426, 1:1000) diluted in blocking solution for 60 min at 4 °C. All antibodies were conjugated with fluorophores except CD75, therefore, the incubation time for the secondary antibody was only done for the CD75 antibody. The cells were then washed and resuspended in 200 μl flow buffer within a FACS tube and were analyzed using a BD FACSVantage SE System at the University of Calgary Flow Cytometry Facility. Gates were drawn based on isotype control. The gating strategy for each marker expression is shown in Supplementary Figs. 6 and 8. The histogram of flow cytometry analysis was generated by FlowJo V10.1.3 software. 10.1.3.

**Whole-mount immunostaining and confocal imaging**. Aliquots of bioreactor-cultured, naïve and primed hPSC aggregates on day 4 post-inoculation were collected into 15 mL conical tubes and were kept in 37 °C for 20–30 min to allow them to settle down to the bottom of the tubes. The media were then removed, and the aggregates were fixed with 2 mL of 4% paraformaldehyde for 30–40 min RT. The aggregates were then washed three times with PBS⁻ containing 0.1% Tween-20 (Sigma–Aldrich, P9416), and permeabilized using PBS⁻ containing 0.25% Triton™

X-100 (Sigma–Aldrich, T8787) for 1 h in RT. Following the washing step, the aggregates were blocked in PBS⁻ solution containing 3% Blocker™ BSA (10X) (Thermo Fisher Scientific, 37525) for 1 h RT. Then, primary antibodies were added to the blocking solution and were incubated with aggregates overnight at 4 °C. The primary antibodies used for whole-mount immunostaining of aggregates were H3K27me3 antibody (Millipore, 07–449, 1:4000), CD75 antibody (Abcam, ab77676, 1 μg for 1 × 10⁶ cells), CD90-PE conjugated antibody (BD Pharmingen™, 561970, 1 μg for 1 × 10⁶ cells), TFE3 antibody (Sigma–Aldrich, HPA023881, 1:500), Oct-3/4 antibody (Santa Cruz Biotechnology, sc-5279, 1:200), KLF4 antibody (Santa Cruz Biotechnology, sc-20691, 1:300), and Stella antibody (Millipore, MAB4388, 1:200). After rinsing three times with PBS⁻ containing 0.1% Tween-20, the aggregates were incubated with secondary antibodies diluted in blocking solution for 1 h RT. The secondary antibodies used for whole-mount immunostaining were Alexa Fluor® 488 (Thermo Fisher Scientific, A21206, 1:1000), Alexa Fluor® 555 (Thermo Fisher Scientific, A-21426, 1:1000), and Alexa Fluor® 546 (Thermo Fisher Scientific, A10036, 1:1000). CD90 antibody was already conjugated with R-phycoerythrin (R-PE) fluorophore, therefore, the incubation time for secondary antibody was only done for the other antibodies. After washing, the aggregates were incubated with Hoechst 33342 (Immunochemistry Technologies, LLC., #639, 1:500) diluted in PBS⁻ for 15 min. Aggregates were washed and then were analyzed by confocal microscopy (Zeiss LSM 880 Confocal Microscope with AiryScan) using 493–630 μm (for H3K27me3, TFE3 and KLF4 coupled to Alexa Fluor® 488), 545–697 μm (for CD75 coupled to Alexa Fluor® 555), 566–691 μm (for CD90 coupled to R-PE), and 550–670 μm (for Oct-3/4 and Stella coupled to Alexa Fluor® 546) filters. Zen Black (Carl Zeiss Microscopy) software was used for Z-stack imaging.

**Image processing and analysis**. Zen Blue (Carl Zeiss Microscopy) software was used for image processing and generating Z-stack videos. Confocal Z-stack videos of the stained aggregates are shown in Supplementary Movies 1–6. Intensity and distribution of H3Kme3 foci in the nuclei of bioreactor-cultured, naïve and primed hPSC aggregates were analyzed using Image J[76]. Six nuclei from each naïve and primed hPSC aggregates were selected (at random) and the intensity and distribution of foci were analyzed by Image J through Plot Profile analysis. Briefly, a symmetric midline was applied in naïve and primed hPSC aggregate's nuclei and the intensity and distribution of H3Kme3 foci were measured on the midline by Plot Profile analysis. A two-dimensional graph was generated from the intensities of foci on the midline displaying H3K27me3 distribution. An arbitrary line was applied on the H3K27me3 distribution graph and the percentage of dots located above that arbitrary line (the percentage of dots located above 40) was measured for both naïve and primed hPSC aggregates.

**Teratoma formation assay**. CB17 SCID mice were purchased from Charles River Company and housed in the animal facility in the Faculty of Medicine, University of Calgary. Animal protocols were performed as approved by the Animal Care Committee of the University of Calgary. The aggregates of naïve and primed hPSCs were collected from bioreactors on day 4 post-inoculation and were dissociated into single cells. Next, 1E6 cells in a total volume of 100 μL PBS⁻ were subcutaneously injected into the hind leg of CB17 SCID mice. At least three were used to assess teratoma formation ability of bioreactor-cultured, naïve and primed hPSCs. Naïve hPSCs formed tumors in mice after 3 weeks (~21d) of injections. The mice, which were injected by naïve hPSCs, were killed and the tumors were dissected and examined by histology. Primed hPSCs formed tumors 8–12 weeks after injection into the mice. A similar approach was done for the histology sections derived from naïve and primed hPSCs. Briefly, the tissue sections were fixed in 4% paraformaldehyde overnight at 4 °C. After dehydration, the tissues were embedded in paraffin (Thermo Fisher Scientific, 503002), and the sections were stained with haematoxylin (Sigma–Aldrich, HHS32) and eosin (Thermo Fisher Scientific, E511) (H&E). The tissues were then examined for different cell or tissue types by transmitted light microscopy.

**In vitro differentiation**. Naïve hPSC aggregates were differentiated into cardio-myocytes, hepatocytes and neural rosettes using the previously published protocols[26,52,53]. The aggregates were collected on day 4 post-inoculation and were either dissociated into single cells using StemPro® Accutase® Cell Dissociation Reagent (Thermo Fisher Scientific, A1110501), or were directly used for in vitro differentiation. For both cardiomyocyte and hepatocyte differentiation, naïve hPSCs as single cells (at the density of 2E5 cells per plate)/or aggregates were cultured on Matrigel® hESC qualified Matrix-coated FluoroDish Cell Culture Dish plates (World Precision Instrument, FD35-100), containing mTeSR™1 or RSeT™ medium until they were 80–90% confluent, and then differentiation procedure was performed using the previously published protocol[52,53]. Mature cardiomyocytes and hepatocytes at day 20 of differentiation, were used for the analysis. For neural rosette differentiation, naïve hPSC aggregates were cultured on Poly-L-Ornithine (Sigma–Aldrich, A-004-C)-coated FluoroDish Cell Culture Dish plates (World Precision Instrument, FD35-100) in DMEM/F12 medium supplemented with 5% Knockout serum replacement (Thermo Fisher Scientific, 10828010), 0.1 mM non-essential amino acids (Thermo Fisher Scientific, 11140050), 0.1 mM 2-Mercaptoethanol (Thermo Fisher Scientific, 21985023), and 1% Penicillin-

Streptomycin (Thermo Fisher Scientific, 15140122) for four days. Further, the aggregates were transferred on to Matrigel-coated plates in Neurobasal™ Medium (Thermo Fisher Scientific, 21103049) supplemented with B27 without Retinoic Acid (Thermo Fisher Scientific, 12587010) and N2 supplement (Thermo Fisher Scientific, 17502048), 0.005% bovine serum albumin (Thermo Fisher Scientific, 15260037), 1 mM sodium pyruvate (Thermo Fisher Scientific, 11360–070) for additional five days before analysis. Differentiated cardiomyocytes and hepatocytes were analyzed by whole-mount immunostaining and confocal imaging using Cardiac Troponin T antibody (Thermo Fisher Scientific, MA5–12960, 5 μg/mL), HNF4-alpha antibody (Abcam, ab92378, 1:100), and CYP3A4 antibody (Thermo Fisher Scientific, MA5-17064, 1:200), respectively. Neural rosettes were analysed using Pax6 antibody (BioLegend, PRB-278P, 1:100).

**Karyotype analysis using spectral karyotyping (SKY).** Naïve and primed hPSC aggregates were collected on day 4 post-inoculation and were cultured for 24 h in adherent conditions. For the comparison, statically cultured, naïve and primed hPSC counterparts were used for the karyotype analysis. The cells were incubated with 0.2 μg/mL KaryoMAX™ Colcemid™ Solution in HBSS (Thermo Fisher Scientific, 15210040) in medium at 37 °C for 1 h and then dissociated and resuspended in 2–3 mL of aqueous 0.068 M KCl (Fisher Scientific, P217) for 15 min at 37 °C. We then fixed the cells with fresh 3:1 methanol: glacial acetic acid, followed by three rinses with fixative solution. We dropped single cells on precleaned slides to spread chromosomes and used the dried slides for SKY analysis. The analysis was performed according to the manufacturer's instructions (ASI, Edingen–Neckarhausen, Baden-Württemberg, Germany) for karyotype analyses on chromosome spreads. 25 and 35 separate metaphase spreads were used for naïve and primed hPSCs respectively to evaluate structural chromosomal aberrations (e.g., balanced/unbalanced translocations and complex rearrangements) in each sample. SKY-Paint DNA-H10 probes (ASI, Edingen–Neckarhausen, Baden–Württemberg, Germany) were used for SKY analysis.

**Statistics and reproducibility.** Exact *n* values for each experiment are defined and described in the main text and figure legends. A one-way analysis of variance (ANOVA) followed by Tukey's multiple comparison test was used for statistical analysis of data related to growth kinetics, RT-qPCR, and flow cytometry. The significance was set at $P < 0.05$ using GraphPad Prism. Replicates are defined as individual passages of naïve and primed hPSCs cultured in stirred suspension bioreactors or under static culture condition. Data are represented as mean ± SEM.

**Reporting summary.** Further information on research design is available in the Nature Research Reporting Summary linked to this article.

## Data availability

The authors declare that all data supporting the findings of this study are available within the article and its supplementary information files or from the corresponding author upon reasonable request. For RNA-seq data, the raw fastq files and raw count table containing the number of transcripts for each sample are available in GEO under accession number GSE144656. Processed RNA-seq data are available in Supplementary Data 1–9 including differentially expressed transcripts in suspension static vs. static culture condition (Supplementary Data 1), enriched canonical pathways in suspension static vs. static culture condition (Supplementary Data 2), differentially expressed transcripts in stirred suspension vs. static culture condition (Supplementary Data 3), enriched canonical pathways in stirred suspension vs. static culture condition (Supplementary Data 4), differentially expressed transcripts in stirred suspension vs. static suspension culture condition (Supplementary Data 5), enriched canonical pathways in stirred suspension vs. static suspension culture condition (Supplementary Data 6), enriched GO terms in static suspension vs. static culture condition (Supplementary Data 7), enriched GO terms in stirred suspension vs. static culture condition (Supplementary Data 8), and enriched GO terms in stirred suspension vs. static suspension culture condition (Supplementary Data 9). Metabolomics raw data are available at the NIH Common Fund's National Metabolomics Data Repository (NMDR) website and the Metabolomics Workbench[77], where it has been assigned the Project ID PR000942.The data can be accessed directly via it's Project https://doi.org/10.21228/M8XM5C. Processed Metabolomics data can be found in Supplementary Data 10

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

## Acknowledgements

We thank Dr. Anne Vaahtokari at the Arnie Charbonneau Cancer Institute, University of Calgary, for her support with confocal imaging. We additionally thank Laurie Kennedy and Yiping Liu at the Flow Cytometry Core Facility for their assistance. We are also thankful to the University of Calgary Center for Health Genomic and Informatics for their support with RNA sequencing. We thank Dr. Ian A. Lewis and Ryan A. Groves from the Calgary Metabolomics Research Facility (CMRF) (supported by the International Microbiome Centre and the Canada Foundation for Innovation CFI-JELF 34986) for their assistance for metabolomics data. I.A.L. is supported by an Alberta Innovates Translational Health Chair. We further thank Dr. Minal Borkar at the Alberta Transplant Institute for assistance with epigenomics data. This project was funded by The Canadian Institutes of Health Research. Leili Rohani was funded by Eyes High Fellowship from the University of Calgary. Breanna Borys was funded through a Vanier Canada Graduate Scholarship. We thank the Stem Cell Network for their generous and continued support in trainee development. The Metabolomic Workbench is supported by NIH grant U2C-DK119886.

## Author contributions

L.R. contributed to concept and design, collection, and assembly of data, data analysis and interpretation, and manuscript writing. B.S.B. contributed to the manuscript's concept, collection and assembly of data, data analysis, interpretation, and manuscript writing. G.R. contributed to collection, assembly of data, and data analysis. P.N. contributed to image analysis, transcriptomic data analysis and interpretation, and heatmap processing. S.L. contributed to cell culture, teratoma formation assay data, and tissue sectioning. A.A.J. contributed to manuscript writing, reviewing, and editing. P.M. contributed to confocal imaging and image analysis. H.H. contributed to authentication and data relevant to the chromosomal stability of the cells used. I.A.L. and R.A.G. contributed to the run and analysis of metabolomics. D.T. contributed to transcriptomic data interpretation, manuscript review, and editing. P.G. contributed to transcriptomic data analysis and interpretation, and IPA pathway analysis. J.W.L. contributed to collection and assembly of data. T.S. contributed to collection and assembly of data. T.D. contributed to collection and assembly of data. M.S.K. contributed to concept and design, manuscript review, editing, and supervision. D.E.R. contributed to concept and design, manuscript review, editing, and supervision.

## Competing interests

The authors declare no competing interests.

## Additional information

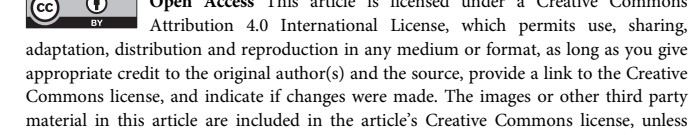

