## [Peer Review File · Communications Biology]

Reviewers' comments:

Reviewer #1 (Remarks to the Author):

The authors claim to have generated a naive hESC line from a well-established primed H9 hESC line using RSeT, and present data comparing the culture of the naive and primed lines, both under stirred suspension or static adherent conditions. They report that naive hESCs cultured with bioreactors are associated with i) superior growth kinetics, ii) upregulation of naive markers and downregulation of primed markers, iii) acquisition of hallmarks of the naive state, iv) retain three germ layer potential and are karyotypically stable. Additionally, the authors suggest that stirred suspension culture itself, through mechanical and/or microenvironmental cues, exerts pressure on the cultured cells to maintain the naive state and, more interestingly, can induce a transitory cell state between primed and naive on primed cells.

The study is generally very innovative and interesting to a broad readership – however few points need to be addressed:

- 1) The data presented confirming naïve state of suspension culture is not convincing. The main manuscript argues with morphology (Fig.1), which is a weak criterion; a more comprehensive molecular characterization is given in Suppl Fig 1A (this key analyses should be shown in the main ms) but is showing a semi-quantitative RT-PCR analysis of XIST, KLF4, SOX2 and NANOG only. However, no controls are given, the XIST band in p4 is significantly lower, and the KLF4 bands very weak. Other “naïve” markers such as ESRRb, REX1, KLF2,5 and primed markers such as OTX2 and ZIC2 are needed employing Q-PCR analyses to support the conclusion of naïve reprogramming. Ideally, these data should be supplemented with single cell data from immunohistochemistry or flow cytometry analyses.
- 2) only one hESC line is represented. Especially since the title references to hPSCs, the authors need to show whether these findings apply also to hiPSCs and/or other hESC lines.
- 3) I disagree with the statement that compared to mPSCs, the manufacturing of conventional hPSCs in stirred suspension bioreactors is cumbersome and inefficient (p.2, l.54). There are several studies published demonstrating efficient hPSC suspension culture, such as Zweigerdt et al 2011 and Kwok et al 2017 (8.9 fold increase per day).
- 4) Since clinical application appears to be a goal, demonstration of a targeted differentiation of these bioreactor-expanded naive hESCs should be demonstrated.
- 5) I feel uncomfortable with the first sentence in the abstract: refers the statement of “pioneering” to previous work or this study? Also, the term “healthier aggregate” must be avoided, since it is not clearly defined.
- 6) Minor comment: it is not immediately clear whether the data presented were generated from only the 50,000 cells/ml inoculation density (as suggested by Fig 2.). If so, this should be clearly stated.
- 7) Few typos (one potentially confusing: "data now shown")

Reviewer #2 (Remarks to the Author):

Rohani et al. describe a study in which the suspension expansion capacity of human pluripotent stem cells is characterized. Cells are treated with a media to convert them to a state in which they express genes associated with the mouse naïve pluripotent state. These cells are then cultured in bioreactors. Rohani et al find that treated cells have shorter doubling times in suspension.

This study would be strengthened with several additional experiments. To be of utility in the bioprocessing industry, continued suspension expansion of these cells should be demonstrated. Serial passaging in suspension for multiple passages will be required for large scale bioreactor seed trains in cell therapy applications, and this proof of concept should be generated showing genetic and karyotypic stability after multiple suspension passages. This should be built into comments that the authors make regarding bioreactor promotion of an intermediate state defined by specific gene expression. The comments on the transitional state induced by bioreactors are not supported by the data presented. A more plausible hypothesis that genes related to suspension conditions (independent of pluripotent state) are observed as differentially expressed (line 250) heterogeneous populations of cells and testing this must be ruled out to make these claims. Claims around the expression of epigenetic associated genes varying according to feed strategy suggest that culture conditions have a significant impact on gene expression (rather than a unique suspension cell state impact – ie all feed conditions are in suspension culture). Similarly, very select genes (specifically focusing on STELLA) are assayed to support these claims. A broader panel of naïve transcription factors needs to be included to support the assertion that bioreactors enhance the naïve-like state. The conclusion regarding culture without ROCK inhibitor is not supported by the data in the paper (line 469).

The primed controls used by the authors cannot be expanded in bioreactor culture. Several literature references cited by the authors describe methods for suspension culture of human pluripotent cell culture. A baseline control in which performance matches expected literature expansion should be included. Similarly, does the method described by the authors work for multiple cell lines, or is this a cell-line specific effect? Suspension culture with additional cell lines would strengthen this paper.

To strengthen this paper further and support bioprocess applications, the authors should demonstrate expansion to densities that are used in industrial suspension bioprocesses or literature stem cell culture processes. The seeding densities and maximum suspension densities in this culture system are an order of magnitude below those demonstrated by others for suspension expansion.

Broadly, this study would be strengthened by illustrating how it builds on existing literature in the field. While the authors claim that little work has been done explicitly comparing naïve to primed in bioreactors (line 74), Lipsitz et al (citation 53) makes this comparison in extensive detail. In fact, several sentences in this paper appear to be taken directly from this publication (lines 54, 82, 94, etc). The authors should carefully paraphrase. Similarly, Gafni et al (citation 16) and Takashima et al (citation 19) describe relevant bioprocessing parameters that the naïve system advances which should be incorporated into the methodology and hypotheses that the authors follow.

In summary, the authors present a study that is one of the first describing the enhancement of human pluripotent stem cell suspension bioprocessing through naïve cell state conversion. For inclusion in this publication, the reviewer advises the authors to clearly position the findings of this paper in terms of how they advance the existing literature and science in the field; to demonstrate that the system is robust across different cell lines, higher seeding densities, and multiple suspension passages; to build out the novelty and story of how suspension culture promotes naïve pluripotency and the naïve state; and to polish up the claims in the text so they are all clearly supported by the data shown in the figures.

Minor comments:

- Line 115 makes a conclusion comparing naïve hPSC to mPSC without data shown. Either show the data or remove this statement
- Review language and style guideline for this journal and phrase introduction accordingly
- Line 51 – define efficiency in terms of routine bioprocess parameters measured
- Line 103 is missing a space
- Line 121 – define cognate primed hPSC

- Line 123 replace as with at

Reviewer #3 (Remarks to the Author):

In this manuscript, Leili and colleagues first show that hPSCs exhibit superior growth in suspension bioreactors compared to their primed counterpart, and they also proved that the bioreactor-cultured naïve hPSCs display quite good naïve features compared to their primed counterparts in terms of gene expression, epigenetic markers, cell surface protein markers and more importantly the teratoma formation ability. It is of great interesting to converting primed to the naïve hPSCs in the bioreactor without the potential need for inhibitors and cytokines, for which are quite important for both clinical and research applications of hPSC. This project was based on their previous work on mouse system and the similar method seems working in the human PSC too, for which and the reviewer thought is quite suitable to published timely to the readers in the stem cell field.

Major concerns:

- 1 Page 5. Line 211, the authors suggest that "a naïve pluripotency-promoting substance is accumulating in the culture medium". Which kind of substance is accumulating? Or it is possible that some primed pluripotent- maintaining substance is reduced?
- 2 How about the metabolism dynamics during the early passages of the primed naïve conversion by the stirred suspension bioreactor method?
- 3 For the detailed molecular mechanism, the reviewer wants to know why the stirred suspension bioreactors can promote the human naïve pluripotency? the difference of signaling dependence or the mechanical force between cell-cell connection comparing to the static culture method?
- 4 How to further improve the stirred suspension bioreactor for a long-term culture of the human Naïve PSC?

The authors would like to thank the reviewers for their careful and detailed review of our manuscript. We believe we have addressed all of the concerns in the table below. We are happy to discuss these if needed.

	Reviewer 1 Major Concerns	Author Response
1	The data presented confirming naïve state of suspension culture is not convincing. The main manuscript argues with morphology (Fig.1), which is a weak criterion; a more comprehensive molecular characterization is given in Suppl Fig 1A (this key analyses should be shown in the main) but is showing a semi-quantitative RT-PCR analysis of XIST, KLF4, SOX2 and NANOG only. However, no controls are given, the XIST band in p4 is significantly lower, and the KLF4 bands very weak. Other “naïve” markers such as ESRRb, REX1, KLF2,5 and primed markers such	The morphological changes in Figure 1 are showing transitional changes in morphology from primed (flat) towards naïve (dome-shaped) hPSC colonies in static culture and not in suspension culture, which is also indicated in the revised manuscript, in the “Results” section “enhanced hPSC bioprocessing”. We have indicated these changes in morphology to mark conversion from primed-to-naïve pluripotent state in static culture followed by several passages of converted naïve hPSCs (P0-P4) on iMEFs to produce established naïve hPSCs before inoculation into suspension bioreactors (Fig. 1 upper panel). Semi-quantitative RT-PCR shown in Supplementary. 1A was performed for the cells from the static culture following establishment of naïve hPSC in static culture (from P0 to P4) to show that the cells expressed naïve pluripotency-related markers before inoculating the bioreactor. We believe this should still be in the supplementary figures as it is not directly dealing with suspension cultured cells. In response to the reviewer request, control H2O is now included in the figure. Since the absence of XIST is a hallmark of naïve pluripotency^{29, 44, 45}, the gradual decline in the expression of XIST from P0 to P4 of naïve hPSCs from static culture supports the presence of naïve populations. Similarly, while the bands for KLF4 are weak, they are absent in primed hPSCs (from static culture), but measurable in naïve hPSCs. The purpose was to show the gradual increase of KLF4 expression from P0 to P4 in statically-cultured naïve hPSCs, in order to confirm established naïve pluripotency before inoculating bioreactors. (Supplementary Fig. 1A). In our original submission, we showed the expression of naïve- (KLF2, KLF4, STELLA) and

as OTX2 and ZIC2 are needed employing Q-PCR analyses to support the conclusion of naïve reprogramming. Ideally, these data should be supplemented with single cell data from immunohistochemistry or flow cytometry analyses.	primed- (CD24) associated genes for H9 cells using RT-qPCR. These genes were repeated for H1. As suggested by the reviewer, additionally, new genes; KLF5 (naïve marker), OTX2, ZIC2, DUSP6 (primed markers) were tested for H1 in the revised manuscript. The results of all these experiments corroborated elevated expression of naïve hPSC-associated genes and reduced expression of primed hPSC-associated transcripts in bioreactor-cultured, naïve hPSCs for H1 and H9 (Fig. 3A, Suppl. Fig. 3A & B). Moreover, to more comprehensively characterize naïve hPSCs in suspension culture, we designed new transcriptomic experiments using RNA-seq for naïve hPSCs cultured under three different conditions: 1) static, 2) static suspension, and 3) stirred suspension culture. Based upon the new transcriptomic experiments, we showed that stirred suspension culture maintains a naïve pluripotency transcriptomic state. The new transcriptomic results supported that naïve pluripotency is maintained in stirred suspension culture in response to mechanical forces (Fig. 3 & 4, Tables 1-3-, Suppl. Fig. 3C). Please see the related “Results” and “Discussion” sections in the revised manuscript. In the original manuscript, we showed single cell flow cytometry analysis data for CD75 (naïve hPSC surface marker) and CD90 (primed hPSC surface marker) for H9. Furthermore, based on recent reports^{24, 49, 50, 51}, we performed single cell flow cytometry analysis for new markers; SUSD2 (naïve pluri-potency), and C24/CD317 (primed pluripotency) for both H1 and H9. The results of all these experiments revealed no remarkable distinction of the naïve pluripotency surface markers (CD75/SUSD2) between bioreactor-cultured, naïve and primed hPSCs. However, all the primed pluripotency cell surface markers (CD90 / CD24 / CD317) exhibited significant downregulation in naïve hPSCs in bioreactor. This is in line with our RNA-seq data which showed a downregulation of primed pluripotency biological pathways. These all suggest that diminished primed pluripotency hallmarks are supportive of naïve
---	--

		pluripotency maintain in the bioreactor (Figs. 7 & 8, Suppl. Figs 5 & 6). Please see the related “Results” and “Discussion” sections in the revised manuscript.
2	Only one hESC line is represented. Especially since the title references to hPSCs, the authors need to show whether these findings apply also to hiPSCs and/or other hESC lines.	We have repeated the experiments with a second hESC line H1. The following experiments were applied to this new H1 line: bioreactor growth kinetics, RT-qPCR, epigenetic regulator expression, metabolomics, flow cytometry, targeted in vitro differentiation, and karyotype analysis. The trends for bioreactor-cultured, naïve hPSCs from the original H9 line held true for the H1 line tested.
3	I disagree with the statement that compared to mPSCs, the manufacturing of conventional hPSCs in stirred suspension bioreactors is cumbersome and inefficient (p.2, 1.54). There are several studies published demonstrating efficient hPSC suspension culture, such as Zweigerdt et al 2011 and Kwok et al 2017 (8.9-fold increase per day).	The words “cumbersome” and “inefficient” were removed from the manuscript, and the following sentence has been adjusted in the manuscript to read, “Compared to mPSCs, the manufacturing of conventional hPSCs in stirred suspension bioreactors has proven more difficulties and resulted in significantly lower fold-expansions ^{14, 15.} ” Please see the related “Introduction” section in the revised manuscript. Both papers mentioned by the reviewer investigate the expansion of human induced pluripotent stem cells. The first paper noted by the reviewer, Zweigerdt et al., achieved a maximum of 6-fold expansion over 4-7 days . The second paper noted by the reviewer Kwok et al., achieved a maximum of 16-fold expansion in 7 days at the 100 mL scale. The statement that they achieved “8.9-fold increase per day” is not mathematically correct based on their own data presented in their manuscript. In both these studies, following 48hrs of culture, full media exchanges were performed. In comparison, in our publications studying mPSC expansion in bioreactors, we have achieved a maximum of 31-fold expansion in 5 days ⁹ , 31-fold expansion in 4 days ⁸ , and 54-fold in 4 days of batch culture ¹⁵ . These significant differences between mouse (31-54x) and human (only 6-16x) are the reason for the statement in the manuscript.
4	Since clinical application appears to be a goal, demonstration of a targeted differentiation of these bioreactor-	In the original manuscript, we have shown the differentiation potential of bioreactor-cultured, naïve hPSCs via teratoma formation for H9 hESC

	expanded naïve hESCs should be demonstrated.	line. We have performed new experiments, performing targeted differentiation of bioreactor-cultured, naïve hPSCs into hepatocytes, beating cardiomyocytes, and neural rosettes for the H1 hESC line (Fig. 10A & B, Suppl. Movie 20).
5	I feel uncomfortable with the first sentence in the abstract: refers the statement of “pioneering” to previous work or this study? Also, the term “healthier aggregate” must be avoided, since it is not clearly defined.	The word “pioneering” and the phrase “healthier aggregate” have been removed from the “Abstract”.
	Reviewer 1 Minor Concerns	Author Response
1	It is not immediately clear whether the data presented were generated from only the 50,000 cells/ml inoculation density (as suggested by Fig 2.). If so, this should be clearly stated.	For all the data the inoculation density was 50,000 cells/ml, otherwise it has been mentioned. We have now included this sentence with the figure captions “The data presented are generated from an inoculation density of 50,000 cells/ml”.
2	Few typos (one potentially confusing: "data now shown")	We have endeavored to find all of the typos and have removed the following sentence from the manuscript “An agitation rate of 120 RPM prevented aggregate formation and 80 RPM caused aggregates with necrotic centres (data are now shown)”.

	Reviewer 2 Major Concerns	Author Response
1	To be of utility in the bioprocessing industry, continued suspension expansion of these cells should be demonstrated. Serial passaging in suspension for multiple passages will be required for large scale bioreactor seed trains in cell therapy applications, and this proof of concept should be generated showing genetic and karyotypic stability after multiple suspension passages. This should be built into comments that the authors make regarding bioreactor promotion of an intermediate state defined by specific gene expression.	Long-term bioreactor expansion of naïve hESCs (via serial passaging) has proven to be a challenge. We believe that a much larger study is required to optimize the culture media and bioprocess conditions for naïve hPSC bioreactor serial passaging. We have devoted a paragraph in the “Discussion” surrounding serial passaging of naïve hPSCs in bioreactor culture. Please see the related paragraph in the revised manuscript. In the original version of the manuscript, we examined karyotype stability with no further structural chromosomal aberrations in bioreactor-cultured, naïve and primed H9 hPSCs. We have repeated the experiment with a second hESC line (H1). The new results confirmed no clonal

		occurrence of structural chromosomal aberrations in bioreactor-cultured, naïve and primed hPSCs following recovery in static condition compared to their original statically-cultured cells (Fig. 10C, Suppl. Fig. 9).
2	The comments on the transitional state induced by bioreactors are not supported by the data presented. A more plausible hypothesis that genes related to suspension conditions (independent of pluripotent state) are observed as differentially expressed (line 250) heterogeneous populations of cells and testing this must be ruled out to make these claims.	To more comprehensively characterize naïve hPSCs in suspension culture, we designed new transcriptomic experiments using RNA-seq for naïve hPSCs cultured under three different conditions: 1) static, 2) static suspension, and 3) stirred suspension culture. According to data from these new experiments, we found: 1) Cell-cell connections alone are not sufficient to maintain naïve pluripotency (based upon the downregulation of Mouse Embryonic Stem Cell Pluripotency pathway in static suspension), 2) Exclusive enrichment of the mechano-sensing HIPPO signalling (regulated by cell-cell connections and mechanical force) in the stirred suspension culture suggesting the importance of a mechanical environment for maintaining naïve pluripotency in the bioreactor, 3) Both static suspension and stirred suspension culture (3D cultures) displayed downregulation of pathways related to cell-cell connection, adhesion and ECM. This supports the hypothesis that common differentially expressed genes in suspension culture are not the specific contributing factors in maintaining naïve pluripotency, and 4) A decrease of lineage-specific and primed pluripotency related pathways in stirred suspension culture, which can be supportive of naïve pluripotency in the bioreactor. These all support the positive influence of bioreactor environment on naïve pluripotency, and that the naïve pluripotency state is maintained in stirred suspension culture in response to mechanical forces (Fig. 3 & 4, Suppl. Tables 3, 6, 9, Suppl. Fig. 3C). Please see the related “Results” and “Discussion” sections in the revised manuscript. In addition to the new RNA-seq analysis, we added the following new experiments to the revised manuscript: RT-qPCR for broader panels of naïve and primed pluripotency associated transcripts (Fig. 3A), metabolomics (Fig. 6), and flow

		cytometry using the recently reported surface markers for naïve and primed hPSCs (Figs. 7 & 8, Suppl. Figs. 5 & 6). The results of all those experiments corroborate that the bioreactor’s mechanical environment support the naïve pluripotency state.
3	Claims around the expression of epigenetic associated genes varying according to feed strategy suggest that culture conditions have a significant impact on gene expression (rather than a unique suspension cell state impact – ie all feed conditions are in suspension culture). Similarly, very select genes (specifically focusing on STELLA) are assayed to support these claims.	In the original manuscript, we showed elevated expression of epigenetics associated genes; DNMT3L, MAGEB2, and PRDM14 in bioreactor-cultured, naïve hPSCs (both batch and fed-batch conditions). In the revised manuscript, we added more new epigenetic regulators; DNMT1, TDG, and EP300 for both H9 and H1 hESCs. For this new comparison, and to maintain consistency, all the bioreactor suspension cultures underwent fed-batch conditions (60% media change 48h post-inoculation). The new tested epigenetic regulators demonstrated no difference between bioreactor-cultured, naïve hPSCs and their primed equivalents. (Fig. 5, Suppl. Fig 4) There is a lack of agreement in the literature surrounding how the expression levels of these epigenetic regulators change with different naïve pluripotency media compositions ^{19, 20, 34, 43}. Considering this, and that some of the epigenetic regulators (DNMT3L, MAGEB2, PRDM14) were highly expressed in the bioreactor, and some were not, we believe that there may be heterogenous populations of naïve cells harboring different epigenetic and naïve pluripotency stages in the bioreactor. However, truly exploring the entity of those cells may require extensive high-throughput analyses such as single cell RNA-seq ²³ and Methyl-Seq target enrichment bisulfite sequencing/DMRs ³⁵ which are currently beyond the scope of this present study. We therefore propose much larger studies for the future work. Please see the related “Results” and “Discussion” sections in the revised manuscript. According to previous studies which showed that naïve and primed hPSCs differ in their metabolome thereby affecting their epigenetic status ^{38, 46}, we included metabolomics analysis for both H1 and H9 hESC lines in the revised manuscript. The dynamics of metabolites revealed robust production

	A broader panel of naïve transcription factors needs to be included to support the assertion that bioreactors enhance the naïve-like state.	of the key naïve pluripotency metabolites, α-KG and glutamate^{47, 48} in naïve hPSCs in the bioreactor (Fig 6). α-KG particularly reported to promote histone/DNA demethylation, helps maintaining naïve pluripotency^{47, 48}. The existence of those metabolites, particularly α-KG (the main metabolite in naïve pluripotency), in the bioreactor can help retain high expression levels of DNA demethylation transcripts such as DNMT3L, PRDM14, and MAGEB2, and PGC related pathways such as PTEN signaling (in our new RNA-seq data). Please see the related “Results” and “Discussion” sections in the revised manuscript. To include a broader panel of transcription factors: we incorporated new RNA-seq experiments (explained in the previous comment), and more markers of naïve (KLF5) and primed pluripotency (OTX2, ZIC2, DUSP6) through RT-qPCR analyses (Fig. 3A, Suppl. Fig. 3A & B). The results of the new analysis showed elevated expression of naïve associated transcripts and diminished expression of primed associated genes in bioreactor-cultured, naïve hPSCs.
4	The conclusion regarding culture without ROCK inhibitor is not supported by the data in the paper (line 469).	We have removed this part from the manuscript.
5	The primed controls used by the authors cannot be expanded in bioreactor culture. Several literature references cited by the authors describe methods for suspension culture of human pluripotent cell culture. A baseline control in which performance matches expected literature expansion should be included.	The successful bioreactor expansion of human primed PSCs is dependent on numerous inputs including cell line and media composition and bioprocess variables such as inoculation method, inoculation density, reactor geometry, agitation rate, oxygen availability, and feeding regime. For the purpose of this study, we intended to investigate a head-to-head comparison of primed vs naïve PSC growth. Therefore, we kept all bioprocess variables constant, with no optimization of these variables for either cell state. Optimizing bioprocess variables mentioned above may indeed improve expansion of both cell states and is something we hope to investigate in our future work. For the particular cell line used in our submitted manuscript (H9), similar bioreactor growth patterns have been noted in the literature - “H9 cell line, which in our hands

		is unable to expand in NutriStem medium and expands only moderately in conventional serum-replacement HES medium (1.5 fold in 1 passage),” ²⁵
6	Does the method described by the authors work for multiple cell lines, or is this a cell-line specific effect? Suspension culture with additional cell lines would strengthen this paper.	We repeated the experiments with a second hESC line H1. The following experiments were applied to this new H1 line: bioreactor growth kinetics, RT-qPCR, epigenetic regulator expression, metabolomics, flow cytometry, targeted in vitro differentiation, and karyotype analysis. The trends for bioreactor-cultured, naïve hPSCs from the original H9 line held true for the H1 line tested.
7	To strengthen this paper further and support bioprocess applications, the authors should demonstrate expansion to densities that are used in industrial suspension bioprocesses or literature stem cell culture processes. The seeding densities and maximum suspension densities in this culture system are an order of magnitude below those demonstrated by others for suspension expansion.	We have cultured naïve hPSCs in 24 and 6-well shaken suspension plates at seeding densities of 1E5 and 2E5 cells/mL (shown in Supplementary Fig. 1C, Supplementary Fig. 2B). We found that up to day 3, the fold-expansion in the shaken suspension wells remains the same as in bioreactor culture (seeded at 5E4 cells/mL). At this point, the expansion plateaus in the well plates, possibly due to the feeding strategy. For this comparison, all cultures underwent a single 50% media exchange on day 3. It is possible that an increased supply of nutrients was required for continued expansion of naïve hPSCs seeded at higher densities in the shaken suspension well plates. For this study we purposefully chose to inoculate bioreactors with a lower seeding density, as it is beneficial to the overall bioprocess design. When protocols call for a larger cell seeding density, cells must first be cultured through a lengthy and resource intensive static expansion phase prior to bioreactor inoculation. If the goal is to culture the cells in bioreactors for clinical or manufacturing purposes (50+ L bioreactors), then extended static culture adds a great deal of time and money to the process as static PSCs require a daily media exchange and frequent passaging. Little bioprocess optimization has been presented even at small scale due to the high cost of PSC growth media and labour requirements¹³. High bioreactor seeding densities are a particular disadvantage when working with PSC which are isolated at very low

		quantities (embryonic) or reprogrammed with extremely low efficiencies (between 0.2% to 1.0% for induced pluripotent stem cells). This means that the smaller the cell quantity required for the manufacturing and expansion phase the better. Other publications studying hPSCs in bioreactors have also used seeding densities of this magnitude ^{6, 30} , and studies comparing hPSC seeding densities have shown reduced inoculation densities to have advantageous growth kinetics, with increased fold-expansions and reduced resources required in bioreactor culture ^{14, 30} .
8	Broadly, this study would be strengthened by illustrating how it builds on existing literature in the field. While the authors claim that little work has been done explicitly comparing naïve to primed in bioreactors (line 74), Lipsitz et al (citation 53) makes this comparison in extensive detail. In fact, several sentences in this paper appear to be taken directly from this publication (lines 54, 82, 94, etc.). The authors should carefully paraphrase. Similarly, Gafni et al (citation 16) and Takashima et al (citation 19) describe relevant bioprocessing parameters that the naïve system advances which should be incorporated into the methodology and hypotheses that the authors follow.	We reworded the “Introduction” based on the results of the new experiments and, considering Lipsitz Y. et al. (PNAS 2018), Gafni O. et al (Nature 2013) and Takashima Y. et al. (Cell. 2014), and some recent reports ^{23, 24} on naïve hPSCs. Please see the related parts in the “Introduction” of the revised manuscript.
9	For inclusion in this publication, the reviewer advises the authors to clearly position the findings of this paper in terms of how they advance the existing literature and science in the field.	We have edited the manuscript, to make sure that our two primary messages are explained clearly: 1) Cell-state conversion from primed-to-naïve pluripotency enhances the biomanufacturing of hPSCs, and 2) A mechanical environment is involved in naïve pluripotency in the bioreactor. The future outcomes of those findings have been added to the manuscript. Please see the related parts in the “Introduction” and “Discussion” of the revised manuscript.
	Reviewer 2 Minor Concerns	Author Response
1	Line 115 makes a conclusion comparing naïve hPSC to mPSC without data	We removed this statement from the manuscript.

	shown. Either show the data or remove this statement.	
2	Review language and style guideline for this journal and phrase introduction accordingly.	We have revised the manuscript according to the journal guidelines.
3	Line 51 – define efficiency in terms of routine bioprocess parameters measured.	The authors removed this sentence from the manuscript.
4	Line 103 is missing a space	Corrected accordingly.
5	Line 121 – define cognate primed hPSC	The word “cognate” changed to “equivalent”.
6	Line 123 replace as with at	We have corrected the sentence. Please see the related part (enhanced hPSC bioprocessing) in the “Results” section.

	Reviewer 3 Major Concerns	Author Response
1	Line 211, the authors suggest that “a naïve pluripotency-promoting substance is accumulating in the culture medium”. Which kind of substance is accumulating? Or it is possible that some primed pluripotent- maintaining substance is reduced?	Yes, as the reviewer has suggested, it is possible that a primed-pluripotent- maintaining substance is reduced. Deciphering the accumulating or declining substances in the culture medium of naïve hPSCs in the bioreactor is something that the authors are planning to study in the near future to optimize long-term expansion and maintenance of naïve hPSCs. However, in the revised manuscript, we characterized naïve hPSCs more comprehensively in the stirred suspension bioreactor, by adding the following experiments: 1) RT-qPCR for more naïve and primed hPSC-associated genes (in addition to the previous markers tested in the original submitted manuscript), 2) RNA-seq for naïve hPSCs cultured under three different conditions; static, static suspension, stirred suspension culture, 3) metabolomics, and 4) flow cytometry using the recently reported specific surface markers for naïve and primed hPSCs. To maintain consistent feeding conditions, all the cell cultures underwent a 60% media exchange 48h post-inoculation. Based upon the transcriptomic (RNA-seq & RT-qPCR) and

		flowcytometry experiments, we showed that naïve pluripotency is maintained in stirred suspension bioreactor through the support from diminishing primed pluripotency hallmarks (transcripts, biological pathways, specific cell surface markers) (Figs. 3, 4, 7, 8). According to Metabolomics data, we found robust production of the key naïve pluripotency metabolites, α-KG and glutamate^{47, 48} in the media of naïve hPSCs in the bioreactor (Fig. 6). These data suggest that both production of naïve pluripotency and reduced primed pluripotency related substances are involved in maintaining naïve pluripotency in the bioreactor. More detailed investigation and much more larger studies are required to detect accumulation or reduction of naïve and/or primed substances in the bioreactor. This is something that we are planning to work on.
2	How about the metabolism dynamics during the early passages of the primed naïve conversion by the stirred suspension bioreactor method?	In the revised manuscript, we incorporated metabolomic analyses for H1 (converting from primed-to-naïve in bioreactor) and H9 (established in static culture) hESC lines to infer metabolism dynamics between naïve and primed hPSCs in the stirred suspension bioreactor (Fig. 6). For this, we used high resolution LC-MS untargeted metabolomics to screen for extracellular metabolites in the media of bioreactor-cultured, naïve and primed hPSCs. Dynamics of metabolites revealed robust production of the key naïve pluripotency metabolites, α-KG and glutamate^{47, 48} in the media of naïve hPSCs in the bioreactor. The main metabolite of naïve pluripotency, α-KG, exhibited robust production in both converting and established naïve hPSCs; while, the dynamics of aspartate, glutamate, lactate and malate were different in converting and established naïve hPSCs. These data indicate that the bioreactor environment help to maintain a naïve pluripotency. However, the development of specific 3D naïve pluripotency media, in concert with additional metabolomic bioprocessing, can further improve the future biomanufacturing of naïve hPSCs. Please see the related parts in “Results” and “Discussion” sections in the revised manuscript.
3	For the detailed molecular mechanism,	To more comprehensively characterize naïve

	the reviewer wants to know why the stirred suspension bioreactors can promote the human naïve pluripotency? the difference of signaling dependence or the mechanical force between cell-cell connection comparing to the static culture method?	hPSCs in suspension culture, we designed new transcriptomic experiments using RNA-seq for naïve hPSCs cultured under three different conditions: 1) static, 2) static suspension, and 3) stirred suspension culture. According to data from these new experiments, we found: 1) Cell-cell connections alone are not sufficient to maintain naïve pluripotency (based upon the downregulation of Mouse Embryonic Stem Cell Pluripotency pathway in static suspension), 2) Exclusive enrichment of the mechano-sensing HIPPO signalling (regulated by cell-cell connections and mechanical force) in the stirred suspension culture suggesting the importance of a mechanical environment for maintaining naïve pluripotency in the bioreactor, 3) Both static suspension and stirred suspension culture (3D cultures) displayed downregulation of pathways related to cell-cell connection, adhesion and ECM. This supports the hypothesis that common differentially expressed genes in suspension culture are not the specific contributing factors in maintaining naïve pluripotency, and 4) A decrease of lineage-specific and primed pluripotency related pathways in stirred suspension culture, which can be supportive of naïve pluripotency in the bioreactor. These all support the positive influence of bioreactor environment on naïve pluripotency, and that the naïve pluripotency state is maintained in stirred suspension culture in response to mechanical forces (Fig. 3 & 4, Suppl. Tables 3, 6, 9, Suppl. Fig. 3C). Please see the related parts in “Results” and “Discussion” sections in the revised manuscript.
4	How to further improve the stirred suspension bioreactor for a long-term culture of the human Naïve PSC?	We have devoted a paragraph in the “Discussion” section to this topic, and the new sentences have been added to the “Discussion” surrounding bioprocess optimization of naïve hPSCs.

Reviewers' comments:

Reviewer #3 (Remarks to the Author):

In their manuscript entitled "Stirred suspension bioreactors maintain naïve pluripotency of human pluripotent stem cells (hPSCs)", Rancourt and colleagues reported naïve hPSCs exhibit superior biomanufacturing characteristics in stirred suspension bioreactors compared to primed hPSCs based on their previous development work (Breanna., 2020, Kehoe.,2010). After demonstrating the naïve hPSCs show greater growth rate and better aggregation and bigger than primed hPSCs aggregates, the authors use RT-PCR and RNA-seq to compare transcriptional differences between naïve and primed hPSCs, providing evidence for the role of bioreactor mechanical environment in the maintenance of naïve pluripotency. They then conduct a series of experiments to determine the epigenetic hallmarks, metabolites, cell surface markers and targeted differentiation of naïve and primed hPSCs. Based on this dataset the authors attempt to prove the importance of bioreactor mechanical environment for naïve pluripotency maintaining. The data presented in this manuscript is the extension of these previous works and looks having significant novelty. It will be suitable candidate for publication by revising the points listed below.

Couple points that need to be clarified to further strength it for publication.

1. In Fig S1C, what are the seeding densities of naïve cells cultured in 24 and 6-well shaken suspension plates and bioreactor-cultured (1E5 or 2E5 cells/mL)? Authors showed that shaken suspension culture in well plates resulted in less uniform aggregate sizes with partly necrotic centers. Please check the expression of necrotic markers and exclude existence of apoptosis.
2. In Fig 2, please check the growth properties of naïve and primed hPSCs in static culture condition to validate the advantages of suspension condition.
3. In Fig S2C-D, in both systems, the aggregates began to fall apart, and cell death occurred. Please provide the data for cell death.
4. By comparing the transcriptomics of naïve hPSCs under stirred suspension and static suspension culture, the results showed enrichments of positive and negative pathways such as HIPPO, PTEM and ERK-MAPK in stirred culture (Figure 4). However, validation experiments such as overexpression or knockdown of critical genes in related pathway are required for supporting the importance of bioreactor mechanical environment. And provide GO analysis for these differential expressed genes to make these RNA-seq data clear.
5. Epigenetic modification factor (DNMT3L, MAGEB2 and PRDM14) transcriptome expression level is not equivalent to DNA or histone epigenetic modification state. And what are the expression levels of other factors involving DNA methylation, such as DNMT3A/B, MBD family. Moreover, it is not clear how many pluripotent genes or lineage-specific genes are regulated by above epigenetic regulators.
6. The concept of epigenetic seems to be misused in Abstract and main text because the experiment related epigenetic in this work is immunofluorescence for H3K27me3.
7. In this manuscript, the bioreactor maintains DNA hypomethylation and X-chromosome reactivation, two features that are representative of naïve hPSCs. However, we didn't not see the data for DNA hypomethylation. So please provide these data for whole genome DNA hypomethylation analysis.
8. In Fig 6, the H1 gradually-converted cells showed a decreasing pattern of glutamate and malate from day 2 to day 6, its level highly increased in established naïve (H9 N) hPSCs compared to their primed counterpart (H9 P). Please discuss the reason.
9. In Fig 7, Authors showed that no significant differences in the expression of cell surface markers of naïve pluripotency CD75 or SUSD2 between naïve and primed hPSCs in the bioreactor. Please discuss the reason. Are there other differential expressed surface markers between naïve and primed hPSCs in the bioreactor. Please combine with RNA-seq data and validate them.

The authors would like to thank the reviewer for their support of our manuscript for publication in *Communications Biology*. We appreciate their careful and detailed review of our manuscript, and we have addressed all of the concerns, detailed in the table below. Please note that given the current pandemic and the restrictions imposed by it on lab access, we were unable to perform additional experiments, but rather expanded our bioinformatics analysis and substantially modified the text.

	Reviewer 3 Concerns	Author Response
1	In Fig S1C, what are the seeding densities of naïve cells cultured in 24 and 6-well shaken suspension plates and bioreactor-cultured (1E5 or 2E5 cells/mL)? Authors showed that shaken suspension culture in well plates resulted in less uniform aggregate sizes with partly necrotic centers. Please check the expression of necrotic markers and exclude existence of apoptosis.	We have changed the wording of the caption to clarify that the 24-well plates were inoculated at 1E5 cells/mL, the 6-well plates at 2E5 cells/mL, and the stirred suspension bioreactor at 5E4 cells/mL. We have changed the text now to discuss the morphology of the D5 aggregates taken from shaken suspension, which may have either necrotic or apoptotic centers (lines 138-140, revised manuscript). We no longer have the cell pellets for these and are unable to check the gene expression of necrotic vs. apoptotic markers.
2	In Fig 2, please check the growth properties of naïve and primed hPSCs in static culture condition to validate the advantages of suspension condition.	Several previous works (including our own)^{6, 7, 13, 31} have demonstrated the growth advantage (fold-expansion) of suspension bioreactors compared to static culture for the expansion of primed hPSCs. Other studies have reported on the advanced growth rate of naïve hPSCs under adherent static cultures^{19, 22}. To our knowledge, our study and Lipsitz et al.²⁶ are the only ones to have investigated the growth of naïve hPSCs in a bioreactor and both use previous works to support the notion that this method is advantageous from a manufacturability point of view. Whereas Lipsitz et al.²⁶ compared growth kinetics of naïve hPSCs in dynamic suspension condition (orbital shakers) and small-scale bioreactor without comparing these to static culture, for this study we focused on understanding the role of the bioreactor mechanical environment on maintaining naïve pluripotency. Our comprehensive characterization of naïve hPSCs via transcriptomic (e.g., enrichment of GO terms for cell cycle, biogenesis) and metabolomic approaches does assume advantages in growth properties using a bioreactor, and in the “Introduction” we support this position (lines 59-67 & 90-95, revised manuscript).

3	In Fig S2C-D, in both systems, the aggregates began to fall apart, and cell death occurred. Please provide the data for cell death.	We did not closely examine mechanisms of cell death but agree that this would have interesting. Instead, we have quantified a decrease in viable cells. In Fig S2D, we have shown “Viable Cells/mL” for first few days of passage two of naïve H1 hPSCs in 100 mL stirred suspension bioreactor. The “Viable Cells/mL” graph itself is an indication/hallmark of cell death. Furthermore, to make these results clearer, in the revised manuscript, we have incorporated the average viabilities of naïve H1 hPSCs at first three days of passage two in 100 mL stirred suspension bioreactor as Supplementary Fig. 2E and removed references to cell death from the manuscript (lines 214-218 & & 615-616, revised manuscript).
4	By comparing the transcriptomics of naïve hPSCs under stirred suspension and static suspension culture, the results showed enrichments of positive and negative pathways such as HIPPO, PTEM and ERK-MAPK in stirred culture (Figure 4). However, validation experiments such as overexpression or knockdown of critical genes in related pathway are required for supporting the importance of bioreactor mechanical environment. And provide GO analysis for these differential expressed genes to make these RNA-seq data clear.	We agree with the reviewer on the importance of elucidating the role for specific pathways, and this is something that we are planning to undertake in future studies. However, these validation experiments are beyond the scope of the present study. We performed GO analysis, and the new data have been added to the revised manuscript (Supplementary Fig. 4; lines 320-346, 357-359, and 662-666, revised manuscript).
5	Epigenetic modification factor (DNMT3L, MAGEB2 and PRDM14) transcriptome expression level is not equivalent to DNA or histone epigenetic modification state. And what are the expression levels of other factors involving DNA methylation, such as DNMT3A/B, MBD family.	We have revised the text and re-worded the related “Results” and “Discussion” sections to specifically refer to “the expression levels of epigenetic regulatory transcripts”, in naïve and primed hPSCs in bioreactor, and reworded references to DNA epigenetic modification state (lines 361-371 & 671-672, revised manuscript). In the revised manuscript, we assessed the expression level of other epigenetic regulatory factors, such as the de novo DNA methyltransferase DNMT3B in both H1 and H9 naïve hPSCs (Supplementary Fig. 5B). This epigenetic regulatory transcript demonstrated no difference between bioreactor-cultured, naïve hPSCs

Moreover, it is not clear how many pluripotent genes or lineage-specific genes are regulated by above epigenetic regulators.	and their primed equivalents for H1, while its expression was upregulated in bioreactor-cultured naïve hPSCs in H9 (Supplementary Fig. 5B). There is a lack of agreement in the literature surrounding how the expression levels of these epigenetic regulators change with different naïve pluripotency media compositions ^{20, 21, 35, 36}. In our case, we used RSeT medium which has already been reported to indicate the intermediate naïve pluripotency state with intermediate expression of epigenetic regulatory transcripts (e.g. DNMT3L, DNMT3B) ³⁵. Considering these, and that some of the epigenetic regulators (DNMT3L, MAGEB2, PRDM14) were highly expressed in the bioreactor, and some were not, as well as discrepancy in the expression level of DNMT3B in naïve hPSCs in bioreactor between H1 and H9, we believe that there may be heterogenous populations of naïve cells harboring different epigenetic and naïve pluripotency stages in the bioreactor. However, truly exploring the entity of those cells may require extensive high-throughput analyses such as single cell RNA-seq ²⁴ and Methyl-Seq target enrichment bisulfite sequencing/DMRs ³⁶, which are currently beyond the scope of this study. We therefore propose much larger studies for future work. Please see lines 371-385 & 407-420 in the “Results” and lines 671-707 in “Discussion” sections of the revised manuscript. The authors are aware of a recent report elegantly demonstrating correlations between lineage markers and epigenetic regulators using single cell RNAseq, and that cells cultured under naïve conditions showed a stronger co-regulatory relationship between lineage markers and epigenetic regulators ²⁴. We are very excited to explore how the bioreactor impacts the epigenetic state of naïve hPSCs to switch between maintaining pluripotency or priming for differentiation however, extensive exploration is something that we plan to perform in our future work through more high-throughput analyses such as scATAC-seq. This may in turn suggest beneficial improvements for long-term expansion of them in bioreactor, for instance, by incorporating specific epigenetic modulators to maintain open chromatin.
---	---

6	The concept of epigenetic seems to be misused in Abstract and main text because the experiment related epigenetic in this work is immunofluorescence for H3K27me3.	We have revised the abstract and main text and changed them to be based on the “expression of epigenetic regulatory transcripts associated to naïve pluripotency” and “hallmarks of X-chromosome reactivation” as two indications of naïve pluripotency, rather than epigenetic-based concept. Please see the “Abstract” and lines 361-371 in “Results”, and 671-685 in “Discussion” sections in revised manuscript.
7	In this manuscript, the bioreactor maintains DNA hypomethylation and X-chromosome reactivation, two that are representative of naïve hPSCs. However, we did not see the data for DNA hypomethylation. So please provide these data for whole genome DNA hypomethylation analysis.	We apologize for the confusion and have revised the text to remove “DNA hypomethylation” and the statement “the bioreactor maintains DNA hypomethylation and X-chromosome reactivation” from the related “Results” section (lines 407-413, revised manuscript).
8	In Fig 6, the H1 gradually-converted cells showed a decreasing pattern of glutamate and malate from day 2 to day 6, its level highly increased in established naïve (H9 N) hPSCs compared to their primed counterpart (H9 P). Please discuss the reason.	We believe that there are three possible reasons for the different pattern of glutamate and malate, which we have now explicitly explained in the “Discussion” section (lines 719-749, revised manuscript). The changing dynamics of these metabolites could reflect: 1) The difference in media change between gradually converting and established naïve hPSCs in the bioreactor; 2) A heterogeneous naïve-primed hPSC population in converting condition in bioreactor could additionally impact the concentration of these metabolites; and/or 3) The media shift in the bioreactor in converting naïve hPSCs may cause extra stress on cells. The process of cell adaptation to the newly changed media might affect cell state condition (e.g. cell cycle) and contribute to a different pattern of these metabolites.
9	In Fig 7, Authors showed that no significant differences in the expression of cell surface markers of naïve pluripotency CD75 or SUSD2 between naïve and primed hPSCs in the bioreactor. Please discuss the reason.	We would like to thank the reviewer for their insightful comment on this issue, as we had hoped to see a clear shift in naïve pluripotency surface markers. However, the discrimination of naïve and primed pluripotency cell surface markers has been controversial, and their expression pattern differs with different naïve pluripotency protocols^{36, 51}, with cells cultured in RSeT media representing an intermediate naïve human pluripotency state^{35, 36, 51}. For instance, Collier AJ, et al. reported that cells in RSeT media displayed downregulation of primed pluripotency surface

	Are there other differential expressed surface markers between naive and primed hPSCs in the bioreactor. Please combine with RNA-seq data and validate them.	markers (e.g. CD24 & CD90), but displayed no upregulation of naïve pluripotency surface markers (e.g. CD75) ⁵¹. This is similar to what we have observed in our results. Furthermore, the wider distribution of naïve pluripotency surface markers in bioreactor suggest the presence of a heterogeneous population of naïve hPSCs in bioreactor that may represent an intermediate state. Please see lines 767-778 in the “Discussion” of the revised manuscript. Some other markers such as F11R and CD7 were introduced by Liu X. et al., and Collier AJ, et al. as naïve cell surface markers, however their expression pattern were also different according to the various naïve pluripotency media composition ^{36,51} (please see our response above). We chose SUSD2 as one of the very recent naïve pluripotency surface markers ⁵³, yet its expression in bioreactor-cultured naïve hPSCs was challenging. In contrast, all of the primed pluripotency surface markers showed remarkable downregulation in our study, as appears to be more consistent in the literature ^{26,36,51}. This downregulation of primed pluripotency surface markers, when combined with our RNAseq data showing a downregulation of primed pluripotency biological pathways (Fig. 4 A & B), supports our conclusion that naïve pluripotency is maintained in stirred suspension bioreactors (lines 759-767, revised manuscript). Our current RNAseq data compared naïve hPSCs cultured under three conditions, and primed hPSCs were not included in this transcriptomic comparison, therefore, direct comparison and validation of specific cell surface markers in bioreactor-cultured naïve and primed hPSCs using our RNAseq data is not possible.
--	--	---

REVIEWERS' COMMENTS:

Reviewer #3 (Remarks to the Author):

In the revised manuscript, the authors made careful response for my concerns mentioned before but performed few experiments, leading to lack of solid evidence for naïve feature and concept stated in the text. So naïve hPSCs statement should be changed into naïve-like hPSCs for publication in Communication Biology if the authors could not perform additional experiments due to COVID-19 pandemic.